

# A New Non-linearity Correction Method for Spectrum from GIIRS onboard Fengyun-4 Satellites and its Preliminary Assessments

Qiang Guo[1,2,3,4], Yuning Liu[4], Xin Wang[1,2,3], and Wen Hui[1,2,3]

[1]National Satellite Meteorological Center (National Centre for Space Weather), Beijing 100081, China
[2]Innovation Center for FengYun Meteorological Satellite (FYSIC), Beijing 100081, China
[3]Key Laboratory of Radiometric Calibration and Validation for Environmental Satellites/Key Laboratory of Space Weather,CMA, Beijing 100081, China
[4]Chinese Academy of Meteorological Science, China Meteorological Administration, Beijing 100081, China

*Correspondence to*: Xin Wang (xinwang@cma.gov.cn)

**Abstract.** Non-linearity (NL) correction is a critical procedure to guarantee the calibration accuracy of a spaceborne sensor to approach a good level (i.e. better than 0.5K). Unfortunately, such a NL correction is still unemployed in spectrum calibration of Geostationary Interferometric InfraRed Sounder (GIIRS) onboard Fengyun-4A (FY-4A) satellite. Different from the classical NL correction method where the NL coefficient is estimated from out-band spectral artifacts in an empirical low-frequency region originally with prelaunch results and updated under in-orbit condition, a new NL correction
method for a spaceborne Fourier transform spectrometer (including GIIRS) is proposed. Particularly, the NL parameter $\mu$ independent of different working conditions (namely the thermal fields from environmental components) can be achieved from laboratory results before launch and directly utilized for in-orbit calibration. Moreover, to overcome the inaccurate linear coefficient from two-point calibration influencing the NL correction, an iteration algorithm is established to make both the linear and the NL coefficients to be converged to their stable values with the relative errors less than 0.5% and 1%
respectively, which is universally suitable for NL correction of both infrared and microwave sensors. By using the onboard internal blackbody (BB) which is identical with the in-orbit calibration, the final calibration accuracy for both all the detectors and all the channels with the proposed NL correction method is validated to be around 0.2-0.3K at an ordinary reference temperature of 305K. Significantly, in the classical method, the relative error of NL parameter immediately transmitting to that of linear one in theory which will introduce some additional errors around 0.1-0.2K for the interfered
radiance inevitably, no longer exists. Moreover, the adopted internal BB with the higher emissivity will produce the better NL correction performance in practice. The proposed NL correction method is scheduled for implementation to GIIRS onboard FY-4A satellite and its successor after modifying their possible spectral response function variations.

## 1 Introduction

The Geostationary Interferometric InfraRed Sounder (GIIRS), the first geostationary Fourier transform spectrometer (FTS),
is onboard Fengyun-4A (FY-4A) satellite to provide high temporal resolution (at the order of $10^1$–$10^2$ minutes) information





on the atmospheric state for numerical weather prediction (NWP) and nowcasting, which is benefit to monitoring and forecasting applications at regional scales (Guo et al., 2021b). Currently, there are two GIIRS-type sensors onboard both FY-4A and Fengyun-4B (FY-4B) satellites where the former and the latter were launched on December 11, 2016 and June 3, 2021 respectively. In general, the two sensors (namely FY-4A/GIIRS and FY-4B/GIIRS) are similar in main spectral

characteristics (i.e. spectral resolution is 0.625cm$^{-1}$, spectral calibration accuracy is 10ppm, and spectral range and channels for midwave infrared (MWIR) band are 1650-2250cm$^{-1}$ and 961, respectively) except that the spectral range of longwave infrared (LWIR) of FY-4B/GIIRS is extended from 700cm$^{-1}$ of FY-4A/GIIRS to 680cm$^{-1}$, where the corresponding spectral channels of FY-4B/GIIRS in LWIR band is increased by 32. Meanwhile, the total detector number of GIIRS for both FY-4A and FY-4B satellites is the same to be 128, the configurations of detector array of which differ with each other (i.e. 32×4 is

for FY-4A/GIIRS and 16×8 for FY-4B/GIIRS). Particularly, compared with FY-4A/GIIRS, both the radiometric and the geometrical characteristics (i.e. sensitivity, radiometric calibration accuracy and spatial resolution) of FY-4B/GIIRS behave significantly better as indicated in Table 1. Such improvements of FY-4B/GIIRS are expected to provide measurements with the higher quality against its predecessor from space. It was partially validated by both domestic and international users that the spectral and radiometric accuracies of the measured spectra from FY-4A/GIIRS V3 algorithm for L1 data show a well-

behaved performance for both LWIR and MWIR bands (Guo et al., 2021b). However, the non-linearity (NL) correction has still not been implemented in the latest V3 algorithm. Therefore, in order to increase the radiometric accuracy further, a new NL correction method which is aimed to carry out the NL processing of GIIRS is proposed in this article.

**Table 1. Main Specifications of LWIR and MWIR bands for GIIRS onboard FY-4A/B satellites**

| Satellite | FY-4A | FY-4B |
|---|---|---|
| Spectral Range | LWIR: 700-1130 cm$^{-1}$  MWIR: 1650-2250 cm$^{-1}$ | LWIR: 680-1130 cm$^{-1}$  MWIR: 1650-2250 cm$^{-1}$ |
| Spectral Resolution | 0.625cm$^{-1}$ | 0.625cm$^{-1}$ |
| Spectral Channels | LWIR: 689   MWIR: 961 | LWIR: 721   MWIR: 961 |
| Number of Detectors | 128: 32×4 | 128: 16×8 |
| Spatial Resolution (@nadir) | LWIR/MWIR: 16 Km | LWIR/MWIR: 12 Km |
| Sensitivity (mW/(m$^2$·sr·cm$^{-1}$)) | LWIR: 0.5-1.1  MWIR: 0.1-0.14 | LWIR: <0.5  MWIR: <0.1 |
| Radiometric Calibration accuracy | 1.5 K | 0.7 K |
| Spectral Calibration accuracy | 10 ppm | 10 ppm |

The NL correction method ordinarily used for most FTS is an approach first developed by the Space Science and Engineering Center at the University of Wisconsin–Madison (UW-SSEC) (Han, 2018; Knuteson et al., 2004a; Qi et al., 2020). Measurements from an onboard FTS are affected by NL diversely in different bands. In particular, for the LWIR and MWIR, the detectors have larger NL contributions to be corrected against those of SWIR which are negligible small without





correction (Qi et al., 2020; Zavyalov et al., 2011). Therefore, the NL correction for LWIR and MWIR bands of detectors is usually considered in most current researches. In a FTS, the NL manifests itself by distortions of the resultant spectrum in the in-band spectral region. It also creates out-band spectral artifacts both in the low-frequency region and at the harmonics of the in-band spectrum (Chase, 1984; Wu et al., 2020). Therefore, the analysis of the spectral range between zero and the lowest detectable wavenumber for the presence of spurious spectral response is an important diagnostic of the polynomial NL response in FTS measurements (Chase, 1984), although such a spurious spectral response is not strictly one-to-one correspondence of NL caused by detector in theory. By looking for nonzero intensity in low frequency regions where the detector response is known to be zero, the final NL coefficient can be obtained (Chase, 1984). The approach has been applied to many interferometers, such as AERI (Knuteson et al., 2004a, b), CrIS (Han, 2018; Han et al., 2013; Taylor et al., 2009; Zavyalov et al., 2011), HIRAS (Qi et al., 2020; Wu et al., 2020), TANSO (Kuze et al., 2012), S-HIS, NAST-I (Revercomb et al., 1998). In addition, there are also other two methods to determine the NL coefficient values, which have been applied to CrIS data. One of the methods uses the external blackbody calibration target (ECT) during prelaunch ground thermal vacuum tests, with the NL coefficient values determined from the spectra when the instrument views the ECT at a set of temperatures. The other one relies on a reference field-of-view (FOV) which has the lowest NL among the other FOVs and derives NL coefficient values for the other FOVs relative to the reference FOV, which can be applied for both prelaunch and in-orbit calibrations (Han, 2018; Tobin et al., 2013; Zavyalov et al., 2011).

In fact, for a traditional broad band infrared sensor, such as MODIS, GOES, VIIRS, MERSI (Datla et al., 2016; Oudrari et al., 2014; Xiong et al., 2003), it is in need to determine and correct the NL response during calibration, particularly for the quadratic contribution of NL. Therefore, the quadratic NL coefficient is calculated in the laboratory calibration before launch and adopted directly for utilization under the in-orbit condition for most infrared sensors. Theoretically, however, both the linearity and the NL terms are affected by the background radiation changes from the environmental components (Guo et al., 2021a), the thermal fields of which consist of different working conditions of a sensor (i.e. GIIRS), so the NL coefficient is inconstant with respect to the linearity response. Meanwhile, for some microwave sensors, to overcome the NL effects on their calibration accuracies greatly, an optimized meth-od is proposed where the receiver gain $g$ and the system NL parameter $\mu$ are introduced in its calibration procedure. It implies that these calibration coefficients can be well-expressed by $g$ and $\mu$, which has been widely used in most microwave sensors, such as MWRI, SSMIS and SSM/I (Yan and Weng, 2008; Yang et al., 2011). Such a NL correction method for microwave sensors can provide some reference for infrared ones, although the linear coefficient calculated directly from measurements is inaccurate without the NL contribution removed.

The NL principle of GIIRS is essentially the same as that of the traditional broad band sensors, except that the band (LWIR and MWIR) of GIIRS is much wider. Therefore, in this study, a new NL correction method is established for both FY-4A/GIIRS and FY-4B/GIIRS to be universal for most onboard infrared FTSs. Particularly, there are two main steps: Firstly, based on NL principle of infrared sensors (including GIIRS), the accurate linear and NL coefficients are calculated by using laboratory results with some external high-accuracy blackbody (BB) after the spectral response function (SRF) of each detector of GIIRS are estimated. Referring to NL correction of microwave sensors, the NL parameter $\mu$ describing the





relationship between the above linear and NL coefficients are further determined. Secondly, an iterative algorithm is proposed to achieve the accurate NL coefficient with both $\mu$ and the inaccurate linear one directly estimated from two-point calibration method, which provide a new and more accurate way for in-orbit NL correction for both infrared and microwave sensors in theory. In Table 2, the main comparisons of NL correction methods for different types of sensors are provided in detail.

**Table 2. Comparison of NL correction methods for different types of sensors.**

| Sensor Type | Hyperspectral Infrared FTS | Wide-band Infrared Sensor | Microwave Sensor |
|---|---|---|---|
| Principle | Correct the NL of target spectrum according to its out-of-band artifacts in the low-frequency caused by NL | Measure NL characteristics of sensor and correct them in calibration procedure | |
| Application | The interferogram is corrected by NL coefficient and then transferred into spectrum, which behaves linear relationship with radiance. | The NL coefficient is obtained with laboratory calibration and considered to be constant in-orbit, while the linear coefficient is achieved by two-point calibration method. | Both the linear and the NL coefficients are determined by using the NL parameter calculated during laboratory calibration as well as the linear coefficient calculated by two-point calibration method. |

## 2. Materials and Methods

### 2.1 The NL Correction Processing

As mentioned in Section 1, the NL correction of GIIRS can be referred from the calibration method of the broad band infrared instruments, where the relationship between the output digital number (DN) and the received radiance (I) is usually expressed by the quadratic NL formula (Datla et al., 2016; Oudrari et al., 2014; Qi et al., 2012; Xiong et al., 2003), namely

$$I = a_2 \cdot DN^2 + a_1 \cdot DN + a_0, \tag{1}$$

where $a_0, a_1, a_2$ are the calibration coefficients. In general, coefficient $a_0$ for an ordinary measurement from target (for example Earth surface, BB and cold space) should be considered due to the influence of the actual dark current as well as the background radiance from instrument itself. However, since the interested radiances from targets are their net ones, I in Eq.(1) is usually achieved by subtracting cold-space radiance from that of target, which means the alternating current (AC) component of target radiance is retained. Under this condition, it is acceptable that $a_0$ coefficient in Eq.(1) is small enough to be negligible in this study.





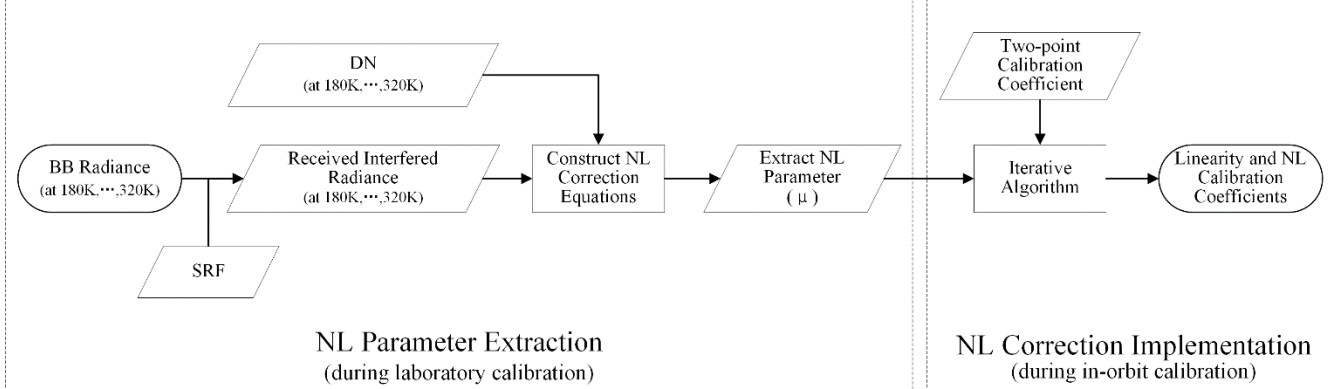

**Figure 1. Processing flow of the proposed NL correction.**

There are two steps of the proposed NL correction, i.e. NL parameter extraction (during laboratory calibration) and NL correction implementation (during in-orbit calibration), as shown in Figure 1.

In the step of NL coefficient extraction, after convolving BB radiance with sensor's SRF, the theoretical interfered radiance (namely interferogram) received by GIIRS can be obtained. Then, by aligning subsample location, the measured interferogram will be converted to the optimized one with the maximal DN, where its zero optical path difference (ZPD) misalignment can approach zero. Then, during laboratory calibration, NL coefficients ($a_2$) can be calculated by fitting the DN with the radiance at different temperatures (180K, …, 320K) by least square method. Finally, the NL parameter $\mu$

describing the relationship between the above linear and NL coefficients are determined for further in-orbit implementation.

In the second step, e.g. NL correction implementation during in-orbit calibration, firstly, the initial linearity calibration coefficient is calculated by the two-point calibration method, the result of which is actually inaccurate without removal of NL contribution. Secondly, an iterative algorithm is adopted to generate the more accurate linear and NL coefficients with both the NL parameter $\mu$ and the initial two-point calibration result.



## 2.2. Principle and Method of NL Correction for Laboratory Calibration

### 2.2.1. Observation Principle of FTS

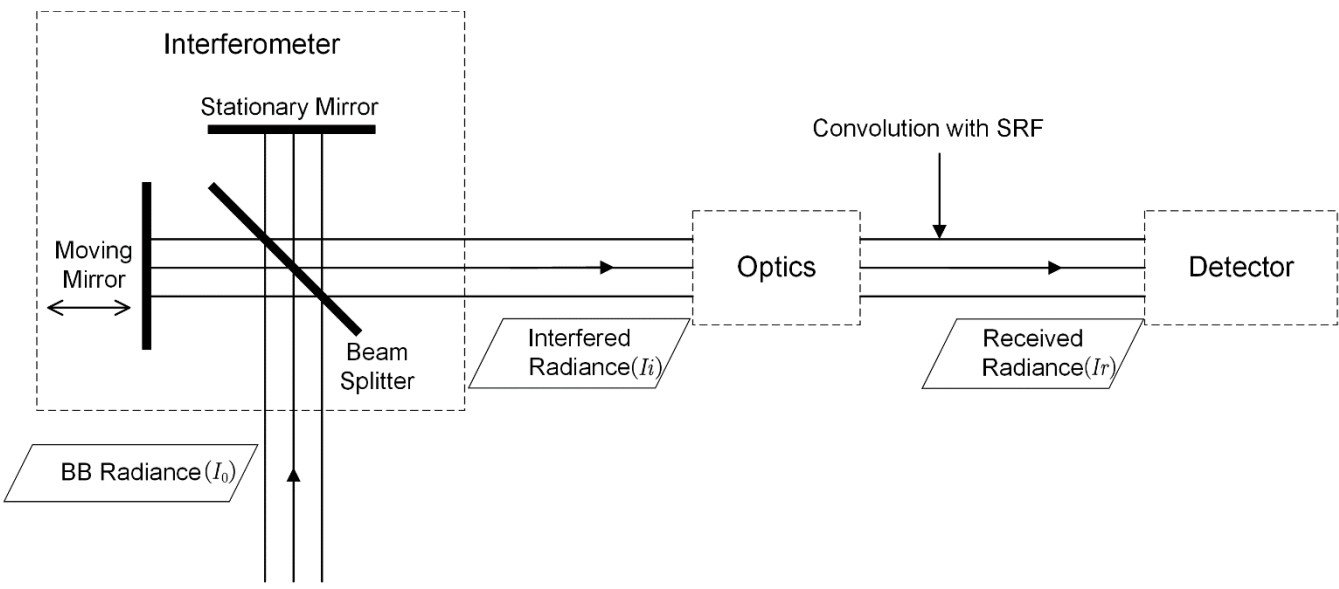

**Figure 2. The simple schematic diagram of Michelson interferometer.**

When implementing NL correction, firstly, it is needed to determine the output DN with its corresponding received radiance by sensor. As shown in Figure 2, the typical Michelson interferometer system (i.e. GIIRS) includes a moving mirror, a stationary mirror, a beam splitter, a detector and other elements, where the moving mirror and the stationary mirror are perpendicular to each other, and their angles against the beam splitter are 45 °. The incident radiance is divided into two parts with exactly the same vibration direction and frequency through the beam splitter. One beam is incident on the stationary mirror and reflected, while the other beam is incident on the moving mirror and reflected. Then they pass through the beam splitter and reach the detector. The moving mirror moves linearly back and forth along the optical axis, which makes the optical path difference (OPD) of the two coherent beams change periodically. Finally, the detector receives an interferogram (or called interfered radiance) with continuous OPD over time. The DNs of the interferogram output from detector are variable with different OPDs. Particularly, in this study, the resulted DN at the location of absolute ZPD is selected for calculation, where the radiation observed by detector can be accurately calculated. Meanwhile, since the observation at absolute ZPD is usually unavailable due to some inevitable subsampling errors, the observed DN value at absolute ZPD can be adjusted from that at the location of approached ZPD.

The radiance of the two interfered beams received by detector is,

$$I_i(T_n) = I_{mov}(T_n) + I_{sta}(T_n) + 2\sqrt{I_{mov}(T_n) \cdot I_{sta}(T_n)} cos(2\pi v x), \tag{2}$$



where the $v$ is wavenumber, $T_n$ is BB temperature, the $x$ is OPD of the two beams, and $I_{mov}$ and $I_{sta}$ are the radiance of the

140 two beams returned by the moving mirror and the stationary mirror and passed through the beam splitter, which are the same

and half of the radiance ($I_0$) incident to the interferometer, that is $I_{mov} = I_{sta} = (1/2)I_0$. As at the ZPD location, where the

OPD $x = 0$ and the radiance is maximal, which means,

$$I_i(T_n) = I_0(T_n)[1 + \cos(2\pi v x)] = 2I_0(T_n),\tag{3}$$

where $I_0$ is the theoretical BB radiance in laboratory calibration, and according to the Planck's blackbody law,

$$I_0(T_n,v) = \frac{c_1 v^3}{e^{c_2 v/T_{n-1}}} \cdot \varepsilon_b(v),\tag{4}$$

where $c_1 = 1.191 \times 10^{-5} W \cdot cm^2$, $c_2 = 1.439 cm \cdot K$, $\varepsilon_b(v)$ is emissivity of the laboratorial BB for different wavenumbers,

and the unit of $I_0$ is $mW/(m^2 \cdot sr \cdot cm^{-1})$.

To calculate the radiance within the whole responsive band of sensor, the theoretical BB radiance needs to have a

convolution with SRF, i.e.

$$I_r(T_n) = 2 \cdot \frac{\int_{v_1}^{v_2} srf(v) \cdot I_0(T_n,v) dv}{\int_{v_1}^{v_2} srf(v) dv},\tag{5}$$

where $I_r$ is the received radiance, $v_1$ and $v_2$ are the beginning and ending wavenumber of the whole band and $srf(v)$ is the

SRF for each wavenumber, which will be given in Subsection 3.1.1. Furthermore, for the term $I_r(T_n)$ in Eq.(5) which is the

theoretical radiance observed by GIIRS at the absolute ZPD location, no correction is required for off-axis effect upon this

term in practice.

2.2.2. Subsample location alignment

After the theoretical radiance received by sensor is obtained, it is important to determine the theoretical ZPD location to get

the maximal value of interferogram with its phase misalignment approaching zero.

For the discretely sampled interferogram values (I(k)) with its ZPD at ($k_0$)th sample location, the discrete integer ($k_0$) is

highly likely to be misaligned against its true ZPD value by a certain subsample-scale offset. In order to remove such a

160 misalignment, I(k) can be firstly oversampled by $\beta$ times (i.e. $\beta$ can be set to be 100 or larger) into $I_\beta(k)$. Then, when

applying the same ZPD detection method on $I_\beta(k)$ as that on I(k), the ZPD location of $I_\beta(k)$, that is, $k_\beta$ can be easily

determined. Hence, $\Delta k_0$ can be given by (Guo et al., 2021b),

$$\Delta k_0 = (k_0 \cdot \beta - k_\beta)/\beta,\tag{6}$$

Moreover, supposing I($\xi$) is the discrete Fourier transform (DFT) of I(k), it means,

$$\begin{cases} I(\xi) = F[I(k)] = \sum_{k=0}^{N-1} I(k) \cdot e^{-j \cdot (\frac{2\pi k}{N}) \cdot \xi}, \xi \in [0, N-1] \\ I(k) = F^{-1}[I(\xi)] = \frac{1}{N} \sum_{\xi=0}^{N-1} I(\xi) \cdot e^{j \cdot (\frac{2\pi \xi}{N}) \cdot k}, k \in [0, N-1] \end{cases},\tag{7}$$





where F[·] and F⁻¹[·] represent DFT and inverse DFT (IDFT) respectively, and N is the sampling number. Particularly, since the off-axis angle (θ) is spanned between the vectors pointing from the location of a detector to the focal point of optics and the main optical axis respectively, the different ones will make the effective optical path differences of detectors quite different with each other even for the same interference pattern introduced by GIIRS. The off-axis effects ultimately result in the different spectral resolutions among individual detectors without corrections. Therefore, according to the principle of a FTS with double-side interferograms (i.e. GIIRS), for both a given $cos(\theta)$ related to individual detector and a required spectral resolution of $\Delta v$, the sampling number (N) is satisfied with

$$\left\{ N = int\left\{ \frac{10000}{cos\theta \cdot \Delta v \cdot \lambda_{laser}} + 0.5 \right\}, \right. \tag{8}$$

where $\theta$ is the off-axis angle, and $cos\theta$ values of each detector for both forward and backward travel of the moving mirror are accurately measured in prelaunch testing. The $\Delta v = 0.625 cm^{-1}$ is a required spectral resolution and the $\lambda_{laser} = 0.85236$ μm is the reference laser wavelength in micron.

According to the properties of DTF, it can also be written as,

$$I(k_0 - \Delta k_0) = F^{-1}\left[\left(e^{-j \cdot \Delta k_0 \cdot \frac{2\pi \xi}{N}}\right) \cdot I(\xi)\right], \tag{9}$$

where I(ξ) is the DFT of I(k). Therefore, any measured interferogram (I(k)) can be converted to have a smaller ZPD misalignment, which is no more than (1/2β) sample location in theory. The symmetry of the aligned interferogram will be significantly improved. The DN before (original) and after (aligned) subsample location alignment are list in Table 3 for some examples. Obviously, different misalignments will generate different aligned DNs compared with their original ones, where the relative errors are around between 1% and 2% with misalignment more than 0.4 samples.

**Table 3. Examples of the DNs before and after subsample location alignment.**

| $\Delta k_0$ | Original DN | Aligned DN |
|---|---|---|
| -0.27 | 4036.69 | 4072.32 |
| 0.39 | 5093.83 | 5191.25 |

### 2.2.3. Calculation Method of NL Coefficients

After the theoretical received radiance ($I_r$) and the maximal DN value ($DN_m$) at absolute ZPD are obtained, the NL coefficient will be solved by using measurements from several external BB sources with different temperatures, which means,

$$\begin{cases} I_r(T_1) = a_2 \cdot DN_m^2(T_1) + a_1 \cdot DN_m(T_1) \\ \quad \dots \\ I_r(T_n) = a_2 \cdot DN_m^2(T_n) + a_1 \cdot DN_m(T_n) \end{cases}, \tag{10}$$

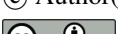



where $T_1$, ..., $T_n$ are different BB temperatures from 180K to 320K. And to remove the influence of background on hot
blackbody (with its temperature higher than 180K) measurements, the $I_r$ and $DN_m$ are subtracted by a cold BB observation
with its temperature around 80K to get both the net radiation and DN with respect to the hot ones, where the radiation from
the cold blackbody itself is neglectable compared with background of sensor.

It should be emphasized that $cos(\theta)$ values for two sweep directions (forward or reverse) differ with each other even for the

same detector, which means off-axis correction for different directions should be performed with the corresponding $cos(\theta)$
in order to uniform the spectral scales for both different detectors and different directions. Theoretically, off-axis correction
aims to determine how many samples of an interferogram should be applied for such a DFT proceeding to obtain the uniform
spectral scales of spectra from targets with different detectors as well as different sweep directions. Actually, the $a_2$ and $a_1$
coefficients in Eq.(10) are utilized to describe the radiometric response characteristics of an individual detector, which is

independent of such a Michelson interferometric optics. After implementing the subsample location alignment for both
forward and reverse sweeps as indicated in subsection 2.2.2, all interferograms from the same detector in both directions are
used to achieve the summed ones with lower noise level. Therefore, in Eq.(10), considering that $I_r(T_n)$ is the inferred
radiance at the absolute ZPD location, it should not been corrected for off-axis effect.

After the laboratory calibration coefficients $a_1$ and $a_2$ are obtained, the NL correction methods for microwave instruments

are referenced, where the relationship between the linearity and the NL coefficients are selected to generate a NL parameter
($\mu$) describing the NL characteristic of sensor. In particular, for a microwave instrument, there is a calibration gain $g$
calculated by two-point calibrations method (Yan and Weng, 2008; Yang et al., 2011), which is equal to the linearity
coefficient $a_1$ in mathematics, i.e.

$$a_1 = g = \frac{I_h - I_c}{DN_h - DN_c}, \tag{11}$$

where $I_h$ and $I_c$ are the radiances of hot and cold BBs, $DN_h$ and $DN_c$ are the corresponding DNs.

Here, the NL calibration coefficient ($a_2$) can be expressed by the gain and the NL parameter ($\mu$) for a microwave sensor,
namely

$$a_2 = \mu \cdot g^2, \tag{12}$$

Therefore, the NL parameter ($\mu$) describing the shape of NL response of sensor in this study can be defined as

$$\mu = a_2/g^2 = a_2/a_1^2, \tag{13}$$

In fact, for a certain detector in either microwave or infrared band, once its NL radiometric responsivity can be given by
Eq.(1), the above NL parameter ($\mu$) may represent the shape characteristic of such a quadratic curve describing its NL
response in mathematics. By calculating the average value of the laboratory calibration coefficients, the final NL parameter
$\mu$ can be obtained, which is theoretically independent of different working conditions (i.e. Normal, Hot and Cold ones)

relevant to the thermal fields from environmental components. Therefore, the μ-parameter method adopted for a microwave





sensor, where the NL responsivity is calculated by using both the NL parameter ($\mu$) and the linearity one, is fully suitable for such an HgCdTe type infrared detector utilized in GIIRS sensor.

### 2.3 A New Iterative Algorithm for In-orbit NL Correction

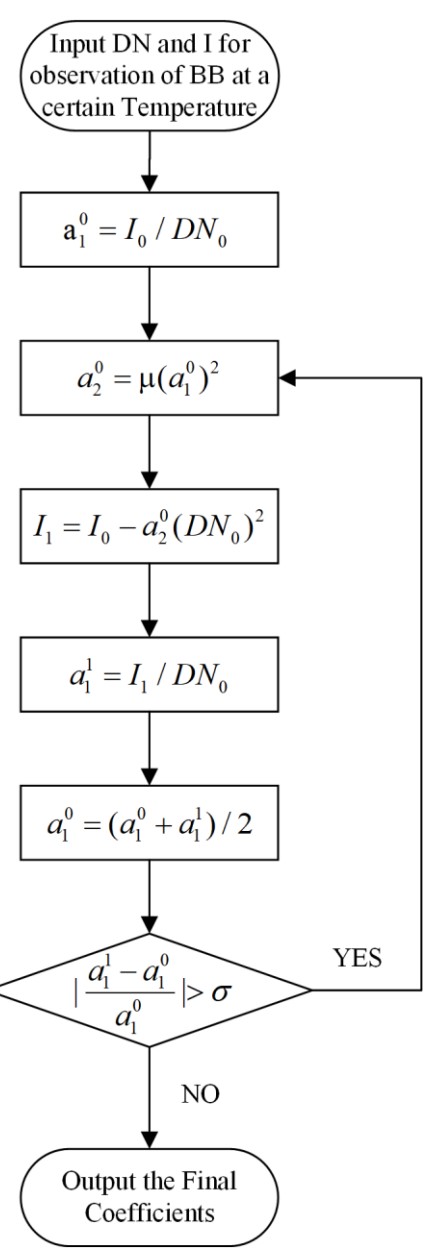

**Figure 3. The iterative algorithm flow of in-orbit NL correction.**





1.  In general, the NL coefficients of most board band infrared sensors from laboratory calibration before launch are directly in-orbit applied. However, it is actually inaccurate to use the laboratory coefficients directly.

2.  In fact, although the NL parameter $\mu$ is introduced to generate a variable NL coefficient $a_2$ together with the linearity one according to Eq.(13), the coefficient of $a_1$ is usually achieved by means of two-point calibration method where the NL influence cannot be removed completely. Therefore, an iterative algorithm is proposed in this study, by dynamically modifying the quadratic NL term ($a_2$), the linearity coefficient is calculated continuously to approach a stable one, and the final accurate linear and NL coefficients can be obtained. The detailed diagram is shown as in Figure 3.

3.  The initial linearity calibration coefficient $a_1^0$ at a certain temperature is obtained.

4.  By using $a_1^0$ and the NL parameter $\mu$ from laboratory results, the initial NL coefficient $a_2^0$ can be generated.

5.  Subtracting the calculated NL contribution from the original radiance $I_0$ to obtain the corrected radiance $I_1$.

6.  The corrected linear coefficient $a_1^1$ can be calculated by $I_1$ and the initial $DN_0$.

7.  The averaged value between $a_1^0$ and $a_1^1$ is used as a updated one of $a_1^0$.

8.  If the relative deviation of the two linearity coefficient ($a_1^0$ and $a_1^1$) is greater than the threshold σ (σ is depended on the required accuracy, i.e. σ can set to be 0.001), a new or updated $a_2$ can be obtained by using the updated $a_1^0$.

9.  Otherwise, when the deviation is less than the threshold σ, the current linearity coefficient $a_1$ is acceptable, while the aimed NL coefficient $a_2$ can be also calculated correspondingly.

**2.4. NL Coefficient Extraction with the Classical Method**

For the classical method for NL correction, the anomalous spectra affected by NL response of a FTS (i.e. GIIRS) at the low-frequency part are used to correct the quadratic NL similar to those in the relevant literatures (Han, 2018; Han et al., 2013; Knuteson et al., 2004a; Tobin et al., 2013). In particular, the out-of-band low-frequency spectrum of 50-450cm$^{-1}$ is empirically selected for calculation.

An interferogram from a detector with a certain of NL characteristics may be related to the ideal interferogram $DN_{ia}$ from a linear detector by the following model (Han, 2018) as

$$DN_{ia} + DN_{id} = (DN_{ma} + DN_{md}) + b_2(DN_{ma} + DN_{md})^2, \tag{14}$$

where $DN_{ia}$ and $DN_{ma}$ are defined as the ideal and measured output AC voltage in volts, $DN_{id}$ and $DN_{md}$ are the direct current (DC) voltage, and $b_2$ is the NL coefficient of classic method. Since the DC item has no any contribution to the interesting spectrum, Eq.(14) can be rewritten as

$$DN_{ia} = (1 + 2b_2 DN_{md})DN_{ma} + b_2 DN_{ma}^2, \tag{15}$$

Implementing DFT on both sides of Eq.(15), we get the corresponding spectra,

$$S_{ia} = (1 + 2b_2 DN_{md})S_{ma} + b_2 S_{ma} \otimes S_{ma}, \tag{16}$$





where $S_{ia}$ is the ideal spectrum, $S_{ma}$ is the measured spectrum, and $S_{ma} \otimes S_{ma}$ is the self-convolution of $S_{ma}$. Assuming that the ideal spectrum of out-of-band spectrum is 0, it can be obtained that,

$$0 = (1 + 2b_2 DN_{md})S_{ma} + b_2 S_{ma} \otimes S_{ma} , \tag{17}$$

Therefore, the NL coefficient $b_2$ can be given by

$$b_2 = \frac{b_2'}{1 - 2b_2' DN_{md}} , \tag{18}$$

where $b_2' = - S_{ma}/S_{ma} \otimes S_{ma}$.

## 3. Results

### 3.1 Introduction to Experimental Data and Instruments

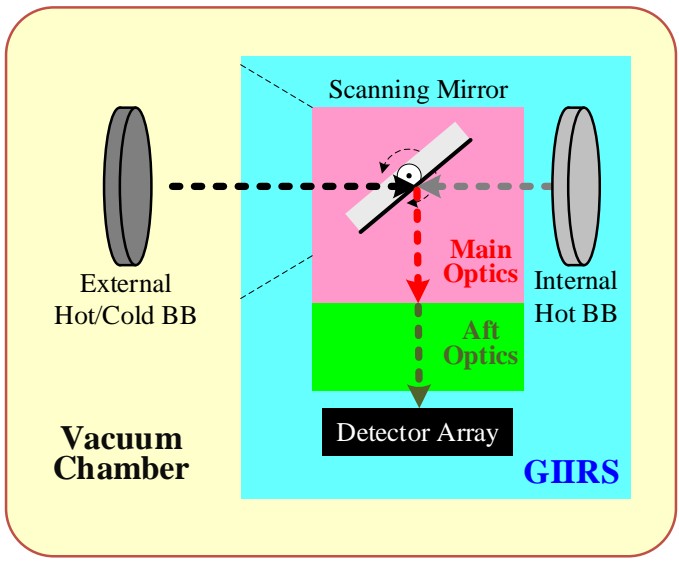

**Figure 4. Sketch of laboratory test in a vacuum chamber for GIIRS**

In order to evaluate the proposed NL correction method more accurately, the experimental data of FY-4B/GIIRS between January 13 and February 11, 2020 (namely its laboratory test/calibration results in a vacuum chamber before launch) were obtained to calculate the NL parameter $\mu$ for different detectors, which are utilized to implement the radiometric calibration together with the corresponding measurements from the internal hot BB target in both in-orbit and in-lab conditions. It

should be mentioned that, as shown in Figure 4, measurements from both the external and the internal BB targets were switched with each other when the scanning mirror is rotated by 90 degrees under a certain of stable situation of GIIRS.





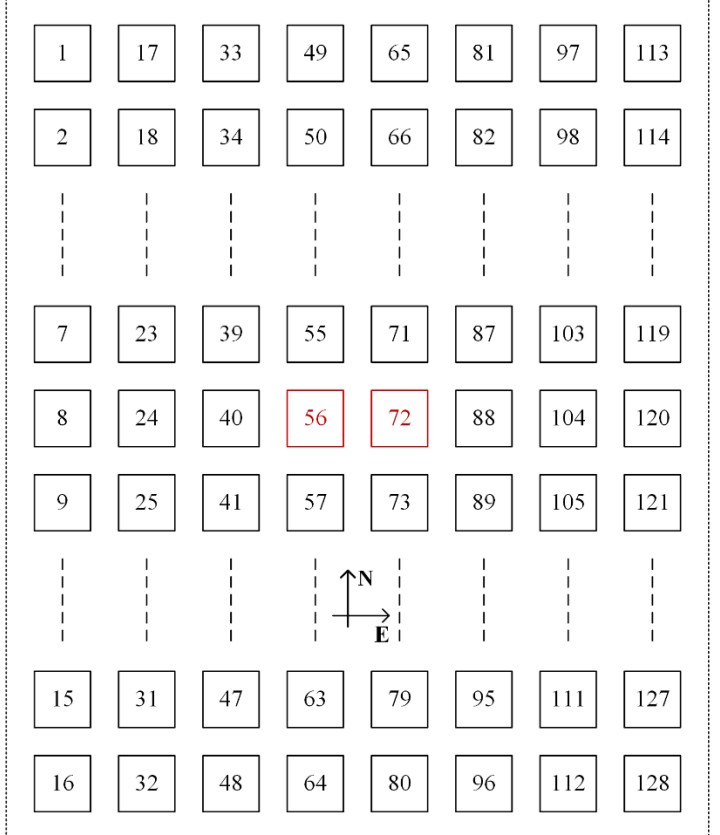

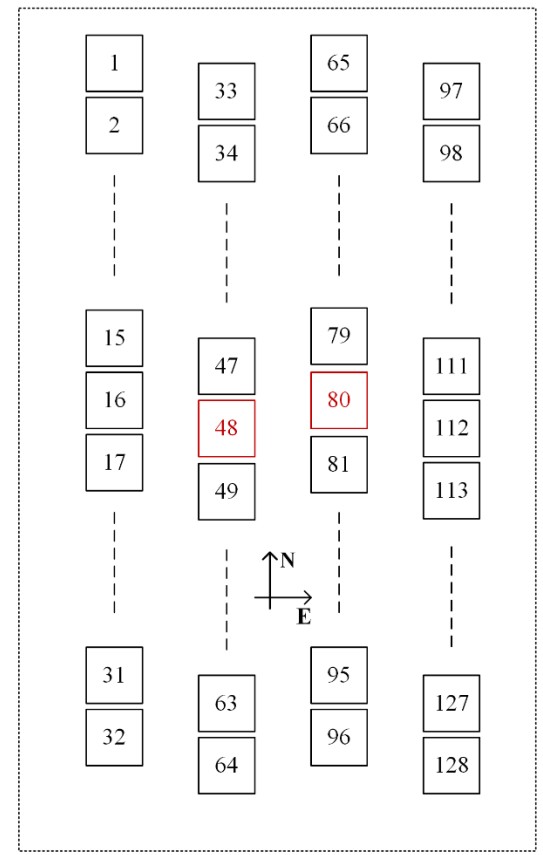

**Figure 5. Layouts of detector array for GIIRS onboard different satellites: (a) FY-4A, and (b) FY-4B.**

Meanwhile, as listed in Table 1, although the total detector number of the LWIR or the MWIR band of both FY-4A/GIIRS
and FY-4B/GIIRS is the same to be 128, the layouts of detector array for the two sensors are quite different as shown in
Figure 5. The main difference is that the detector number for one column in North-South direction has been decreased from
32 for FY-4A to 16 for FY-4B, while the column number in West-East direction has been increased correspondingly from 4
for FY-4A to 8 for FY-4B. Such a change will benefit to reduce the spectral inconsistencies among different detectors due to
their different off-axis angles caused by detector array itself in theory. Particularly, the detectors (marked 48 and 80 in red)
near the central field-of-view (FOV) for FY-4A are also transformed to others (marked 56 and 72 in red) for FY-4B.

During the in-lab test, the calibrating procedures of FY-4B/GIIRS were scheduled to be done under three situations for main
optics (including the scanning and the different reflective mirrors), namely the normal, the cold and the hot ones to assess the
possible variations of radiometric response of GIIRS under different situations. However, for each situation, the aft optics
(including interferometer) were maintained to their optimal temperature conditions (i.e. around 200K for interferometer and
285 65K~75K for the optical assembles related to detectors). Particularly, the approximate temperature ranges of -10 ~ -15
degree, 0 ~ 5 degree and 10 ~ 15 degree are referred to the Cold, the Normal and the Hot situations, respectively.





### 3.2. NL Coefficient and Parameter Extraction with the Proposed Method

#### 3.2.1. Calculation of SRF

SRF of a sensor (i.e. GIIRS) generally refers to the ratio of the received radiation relative to the incident radiation at each
wavenumber. In this study, the SRF of the broad band of GIIRS can be obtained by the laboratory calibration data. The SRF of each wavenumber can be given by

$$srf(v) = S(v)/I_0(v), \tag{19}$$

where $S(v)$ is the DN of the whole band spectrum received by detector, which is calculated from the DFT of the interferogram, and $I_0(v)$ is the theoretical BB radiance incident to the interferometer (GIIRS). Theoretically, the SRF of a certain sensor is an invariable function without any external influencing factors (i.e. irradiation from space), which is independent of external BB source with different temperatures for measurement. However, during the real calculations, the SRF derived from measurements of BB at the lower temperature is inaccurate due to the relative lower signal-to-noise ratio. Therefore, in practice, instead of calculating the SRF with all measurements of external blackbodies with different temperatures from the laboratory calibration, only the mean value of those from the high-temperature (i.e. higher than 290K) ones is selected to estimate different SRFs of individual detectors of GIIRS, the final ones of which are normalized as shown in Figure 6.

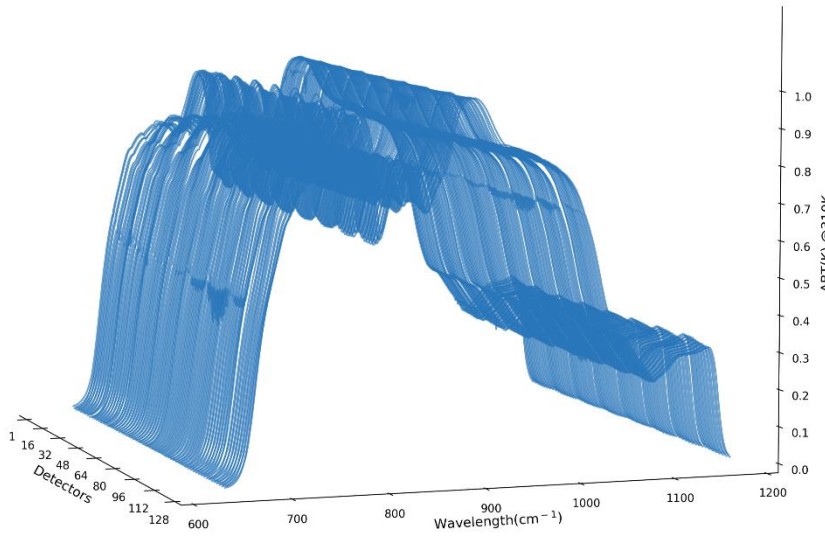

**Figure 6. SRFs for all detectors of FY-4B/GIIRS for LWIR band.**

In general, the spectral dependencies of emissivity of both the external and the internal BB targets are almost identical with each other. Particularly, for the LWIR band (680-1130 $cm^{-1}$), the emissivity for individual channel increases gradually from 0.980 to 0.990 with its wavenumber increasing from $680cm^{-1}$ to $964cm^{-1}$, and then remains slightly variable around 0.990 for





the rest channels with the larger wavenumbers in LWIR one. However, for the MWIR band (1650-2250 cm$^{-1}$), the emissivities of all channels behave poorly varying between 0.955 and 0.970, most (1716-2250 cm$^{-1}$) of which are even worsened to the range between 0.955 and 0.960.

Despite that the nominal band of FY-4B/GIIRS for observation is between 700 and 1130 cm$^{-1}$, it can be seen from Figure 6 that the practical band in which the radiation from targets can be viewed by GIIRS is wider than its nominal one. In order to calculate the radiance more accurate in this study, the wavenumber range of 640-1170 cm$^{-1}$ is chosen as the practical one, while the relative SRF of wavenumber either less than 640 cm$^{-1}$ or greater than 1170 cm$^{-1}$ is around or even less than 0.01, which is small enough to be ignored. Meanwhile, in general, the SRFs of individual detectors of FY-4B/GIIRS approach

with each other, which implies that the spectral responsive characteristics for all the detectors are almost the same at least within such a wide band above.

### 3.2.2. NL Coefficient and Parameter from Laboratory Results

All the three environmental tests (i.e. Cold, Normal and Hot) are adopted to implement the complete calibration procedures before launch. The data selected are the laboratory BB-view measurements with their temperatures between 180K and 320 K

for all the detectors of FY-4B/GIIRS. When using a total of 21 groups of observations with different temperatures within the above range for the external BB for calculation of individual detector, it is found that there is big error of measurements when BB temperature is around 250 K, the exact reason of which is still unknown and have been removed when the final fitting curve with a quadratic item is obtained. Meanwhile, the external BB views at seven different BB temperatures (270-310K) are utilized to assess the calibration accuracy of the proposed NL correction method.

In practice, since the external BB views at around 250K are invalid, the practically utilized measurements from the external BB are exactly with 20 different temperatures. Here, there are at least 3 external BB views at different temperatures to be required to carry out one calculation of the linear and NL coefficients (i.e. $a_1$ and $a_2$). Therefore, the total possible temperature combination number for these calculations is $\sum_{x=3}^{20} C_{20}^x = 1048544$ (Notes: $C_{20}^x$ represents combination of x out of 20), the huge amounts of which guarantees the reliability and stability for the statistical results of both $a_1$ and $a_2$

coefficients. To illustrate the different distributions of both $a_1$ and $a_2$ for different detectors, particularly for those located in different positions of GIIRS's FOV, two typical detectors labelled 56[th] and 96[th], which are located near the central and the marginal positions respectively as shown in subfigure 5(b), are selected. The distributions of two parameters ($a_1$ and $a_2$) with different measurements under the normal situation are provided in Figure 7.





(a)

(b)

(c)

(d)

**Figure 7. Typical distribution diagrams of calibration coefficients for two detectors under the normal situation: (a) the linearity coefficient ($a_1$) of 56th ; (b) the NL coefficient ($a_2$) of 56th ; (c) the linearity coefficient ($a_1$) of 96th ; (d) the NL coefficient ($a_2$) of 96th .**





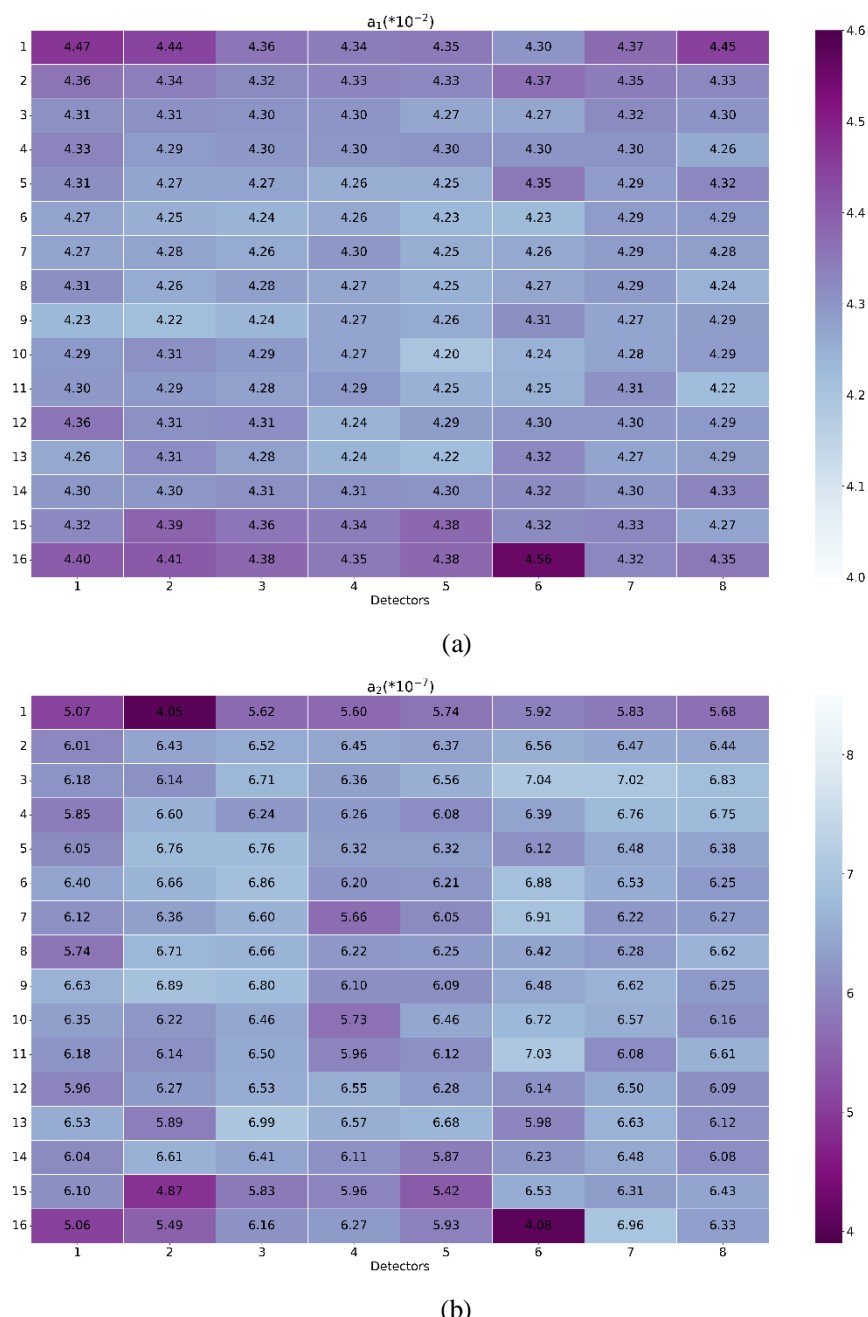

**Figure 8. Mean values of the linearity and the NL coefficients for all detectors of FY-4B/GIIRS under the normal situation: (a) the linearity coefficient ($a_1$) ; (b) the NL coefficient ($a_2$).**

As indicated in Figure 7, both the linearity and the NL coefficients for the two typical detectors are almost satisfied with

normal distribution, and their averaged linear coefficients ($a_1$) are $4.27 \times 10^{-2}$ (56th) and $4.56 \times 10^{-2}$ (96th) respectively, while those of the quadratic NL coefficients ($a_2$) are $6.22 \times 10^{-7}$ (56th) and $4.08 \times 10^{-7}$ (96th) respectively. Moreover, the mean



values of two coefficients ($a_1$ and $a_2$) for all the detectors of FY-4B/GIIRS are also provided and filled in different colors as shown in Figure 8. In general, the values of linearity coefficient ($a_1$) for the central detectors are a certain smaller than those for the marginal ones, the maximal relative differences of which is almost no less than 10%. However, the values of NL

coefficient ($a_2$) for the central detectors are significantly bigger than those for the marginal ones by about 50%, the main reason of which is possibly caused by the overestimated linearity coefficients of the marginal ones.

As an example, according to Eq.(13), the NL parameter of GIIRS ($\mu$) of the central detector ($56^{th}$) under the normal situation is $3.41 \times 10^{-4}$. Meanwhile, the NL parameters of GIIRS ($\mu$) under the cold and hot situations are $3.32 \times 10^{-4}$ and $3.49 \times 10^{-4}$, respectively. Thus, the average value of $\mu$ for $56^{th}$ detector under different working conditions is $3.41 \times 10^{-4}$, and the relative

error of $\mu$ among the three different ones is less than 3%. Such a result indicates that $\mu$ can be regarded as a parameter that can characterize the NL response characteristics of GIIRS, which is independent of different situations, particularly for different temperature configurations of main optics. In this sense, the NL parameter ($\mu$) of each detector of GIIRS, as shown in Figure 9, can represent the mean values for different situations. Similarly, compared with detectors near the central positions of FOV, the NL parameters ($\mu$) of the marginal ones are apparently underestimated by around 50% against those of

central ones, which is also mainly induced by their bigger linearity coefficients. In fact, due to the relative lower optical efficiency at the locations near the marginal areas of FOV, the linearity coefficients which are usually the inverse of responsivities behave bigger than those of the central ones. It implies that the radiometric responsivities of the marginal detectors are generally lower, which can further lead to the smaller NL parameters ($\mu$) even for the same detectors.

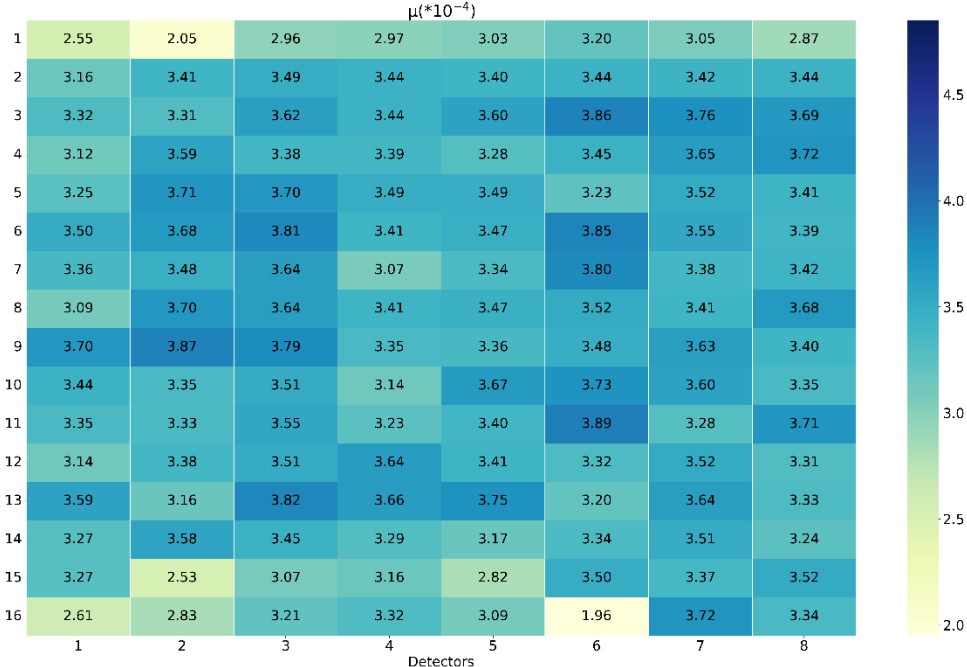

**Figure 9. NL parameter ($\mu$) for all the detectors of FY-4B/GIIRS**





### 3.3. Preliminary assessments of different NL Correction Implementations with Laboratory Results

#### 3.3.1 Performance comparison among three different NL correction methods

To evaluate the real performances of three different NL correction methods (i.e. the proposed one with iteration, the proposed one without iteration and the classical NL one) during the laboratory calibration procedure, the ordinary two-point

calibration mode (i.e. hot point for external hot BB target and cold point for deep space one) is adopted where both the external hot BB with its temperature at 270K, 280K, 290K, 295K, 300K, 305K and 310K respectively and the external cold one with its temperature at 80K (note: regarded as the deep space target in infrared band) are selected to achieve two goals: One is to calculate the NL parameters ($\mu$) with the proposed method described in subsection 2.2 for all the detectors of GIIRS, which are adopted together with measurements of the internal hot BB target to carry out a practical calibration

procedure with the new developed iterative algorithm; The other is to provide some references (i.e. the net radiance from the external hot BB target by subtracting the cold BB observation from the hot external BB one) to assess the calibration performance, which are of the highest accuracy for evaluation. It should be mentioned that such a practical calibration above with the internal hot BB target with the derived NL parameters is fully identical with that under in-orbit condition. Particularly, in this laboratory test of FY-4B/GIIRS, the temperature of the internal hot BB target can be set to 300K, 305K,

310K, 315K and 320K as required. Therefore, for the proposed NL correction method, the related calibration coefficients and iteration information are also provided in Table 4.

For the classical method, after the NL coefficient ($b_2$) is obtained, the interferogram (namely output DN from GIIRS when observing interfered radiance) can be NL corrected and the spectrum after DFT can be linearly calibrated with the internal hot BB target channel-by-channel further. However, the NL coefficients of classical method ($b_2$ for interferogram and

linearity coefficient for each channel) are different from those of the proposed one and cannot be compared with each other directly. Therefore, some additional derivations are included as shown in Eq.(20), the results of which can help to compare the calibration coefficients between the classical and the proposed ones. Since the ideal DN ($DN_{ia}$) given by Eq.(16) has a linear relationship with incident radiance, the calibration equation between incident radiance and the output DN is satisfied with,

$$I_r = c_1 \cdot DN_{ia} = [c_1(1 + 2b_2 DN_{ma})] \cdot DN_{ma} + (c_1 b_2) \cdot DN_{ma}^2 \triangleq a'_1 \cdot DN_{ma} + a'_2 \cdot DN_{ma}^2, \qquad (20)$$

where $I_r$ is the theoretical value of interfered net radiance for the internal hot BB target at different temperatures, $c_1$ is the linearity coefficient calculated by two-point (blackbody and cold space) calibration method and $b_2$ is considered to be constant (Han, 2018). The resulted linearity and NL calibration coefficients are list in Table 4 with respect to the internal hot BB target at different temperatures between 300K and 320K in 5-degree interval.

Moreover, the difference between the actual brightness temperature (BT) and the calibrated one from both the classical and the proposed methods respectively is used to represent the calibration accuracy for the interfered radiance within a wide band (i.e. 640-1170 cm$^{-1}$). Here, the averaged absolute difference in BT at a reference temperature (note: 305K in usual) is,



$$\Delta\mathrm{BT}(T_i) = |BT_{cal}(T_i, T_r) - BT_a(T_i, T_r)| , \qquad (21)$$

where $T_i$ is the temperature of the internal hot BB for calibration, $BT_{cal}(T_i, T_r)$ is the calculated BT of the referenced external
hot BB target at different temperatures $(T_r)$ for calibration, while $BT_a(T_i, T_r)$ is the actual BT with respect to the referenced
one during laboratory calibration. Thus, the calibration results, including the linearity, the NL coefficients, the calibrated BT
difference ($\Delta$BT) at the referenced 305K (i.e. $T_r = 305K$), and the NL parameter $\mu$ for the classical method with internal hot
BB at different temperatures, which is utilized to implement a practical in-orbit calibration, are list in Table 4. Particularly,
according to Eqs.(1), (13) and (20), the deduced NL parameter ($\mu$) with internal hot BB target different temperatures for
classical method is estimated for comparison.

**Table 4. Main results comparison among three different NL correction methods**

| | | FOV-56 ($\mu=3.41 \times 10^{-4}$) | | | | | FOV-96 ($\mu=1.96 \times 10^{-4}$) | | | | |
|---|---|---|---|---|---|---|---|---|---|---|---|
| | Internal BB Temperature (K) | 300 | 305 | 310 | 315 | 320 | 300 | 305 | 310 | 315 | 320 |
| Proposed Method with Iteration | Linear Coefficient ($\times 10^{-2}$) | 4.2116 | 4.2055 | 4.1891 | 4.1779 | 4.1691 | 4.4686 | 4.4607 | 4.4452 | 4.4327 | 4.4224 |
| | NL Coefficient ($\times 10^{-7}$) | 6.0544 | 6.0366 | 5.9899 | 5.9579 | 5.9328 | 3.9159 | 3.9022 | 3.8751 | 3.8534 | 3.8355 |
| | $\Delta$BT(K) | 0.3544 | 0.3191 | 0.3255 | 0.3329 | 0.3294 | 0.5004 | 0.4851 | 0.5542 | 0.5905 | 0.5821 |
| Proposed Method without Iteration | Linear Coefficient ($\times 10^{-2}$) | 4.3260 | 4.3489 | 4.3643 | 4.3807 | 4.4029 | 4.4760 | 4.4989 | 4.5143 | 4.5308 | 4.5529 |
| | NL Coefficient ($\times 10^{-7}$) | 6.3877 | 6.4556 | 6.5012 | 6.5503 | 6.6168 | 6.8384 | 6.9086 | 6.9558 | 7.0066 | 7.0754 |
| | $\Delta$BT(K) | 0.7909 | 1.0139 | 1.1964 | 1.3298 | 1.5769 | 0.9921 | 1.2347 | 1.3971 | 1.5317 | 1.7564 |
| Classical Method | $b_2(\times 10^{-6})$ | 1.6380 | 1.4279 | 1.2748 | 1.1397 | 1.0227 | 3.1416 | 2.8762 | 2.3274 | 2.1396 | 2.0344 |
| | Linear Coefficient ($\times 10^{-2}$) | 4.4875 | 4.5040 | 4.5094 | 4.5207 | 4.5349 | 4.5954 | 4.6016 | 4.6078 | 4.6099 | 4.6128 |
| | NL Coefficient ($\times 10^{-7}$) | 0.7158 | 0.6284 | 0.5630 | 0.5057 | 0.4561 | 1.3721 | 1.2631 | 1.0324 | 0.9525 | 0.9077 |
| | NL parameter $\mu$ ($\times 10^{-5}$) | 3.5545 | 3.0975 | 2.7689 | 2.4746 | 2.2179 | 6.4971 | 5.9651 | 4.8628 | 4.4819 | 4.2659 |
| | $\Delta$BT(K) | 0.3978 | 0.4157 | 0.4326 | 0.4492 | 0.4642 | 0.7995 | 0.7172 | 0.7362 | 0.6782 | 0.6984 |





For the normal situation of FY-4B/GIIRS in laboratory calibration, by using measurements of the internal hot BB target with different temperatures (i.e. 300K, 305K, 310K, 315K and 300K), the accurate calibration results (including linearity and NL coefficients, NL parameter and ΔBT) of two typical detectors (i.e. $56^{th}$ for the central one and $96^{th}$ for the marginal one) are

quantitatively analyzed in three different NL correction methods. Therefore, based on Table 4, several preliminary conclusions can be drawn. Firstly, for the proposed method with iteration, the linear coefficients ($a_1$, the most important contributor to calibration accuracy, their mean values for both $56^{th}$ and $96^{th}$ detectors are 4.1906 and 4.4459, respectively) with the internal hot BB target at different temperatures from 300K to 320K vibrate slightly around their mean values, the maximal relative error of which is less than 0.5%. At the same time, the derived NL coefficients ($a_2$, their mean values for

both $56^{th}$ and $96^{th}$ detectors are 5.9943 and 3.8764, respectively) also behave well with the maximal relative error around 1%. Thereafter, the corresponding BT differences (ΔBT) are generally 0.3~0.4K for $56^{th}$ detector and 0.5~0.6K for $96^{th}$ one. Secondly, however, for the proposed method without iteration, the linear coefficients become bigger and bigger with increasing of the internal BB temperature for calibration for both the two detectors ($56^{th}$ and $96^{th}$ ones), the main reason of which is the more NL influence on $a_1$ from the higher-temperature BB for calibration, and make the NL coefficients ($a_2$) to

be enlarged further with a constant $\mu$ parameter. Without iteration, the final ΔBT values of this method become too big (around 1.6~1.8K for 320K-BB) to be acceptable. Thirdly, for the classical method, the situations are relatively complex. In particular, since $b_2$ is the NL coefficient to describe the NL relationship between the measured DN and the ideal one, it should remain nearly unchanged with respect to the internal BB target for calibration with different temperatures at least in theory. However, as list in Table 4, $b_2$ values are significantly dependent of BB temperature for calibration, namely the

bigger $b_2$ is related to the lower temperature of internal BB. Such results imply that the derived $b_2$ values from the classical method are inaccurate. According to Eq.(20), when the two-point method is adopted to calibrate the interfered radiance or interferogram, the gradually increased linear coefficients ($a_1'$) and decreased NL coefficients ($a_2'$) are inevitable as given in Table 4. And the corresponding ΔBTs of the classical method are a bit larger than those of the proposed one with iteration by 0.1-0.2K. From the perspective of NL correction, the NL characteristics of GIIRS are underestimated by the classical method,

the averaged $\bar{\mu}$ values (i.e. $2.82 \times 10^{-5}$ for $56^{th}$ and $5.21 \times 10^{-5}$ for $96^{th}$) of which are one or half order of magnitude smaller than their true ones (i.e. $3.41 \times 10^{-4}$ for $56^{th}$ and $1.96 \times 10^{-4}$ for $96^{th}$).





### 3.3.2 Preliminary assessments of the proposed NL correction method

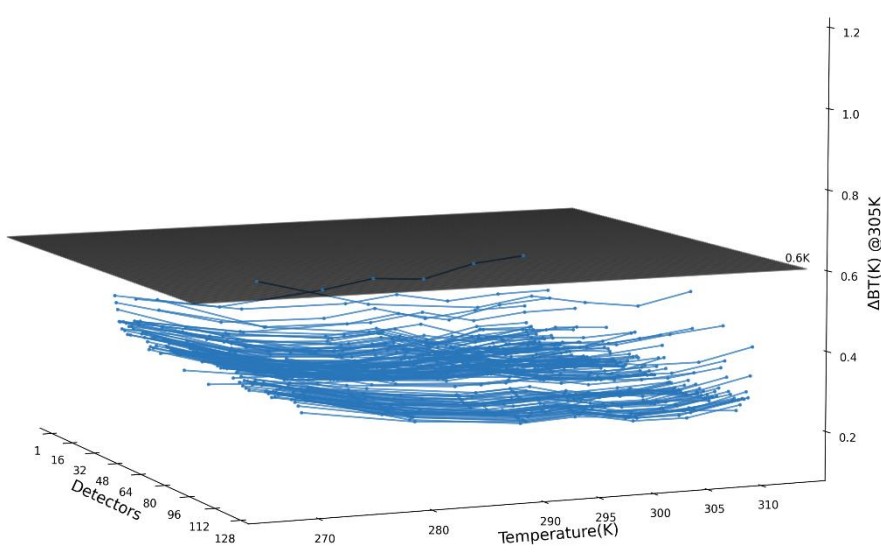

**Figure 10. ΔBT of interfered radiance for all the detectors of FY-4B/GIIRS**

Based on subsection 3.3.1, since the ordinary temperature of the internal hot BB target is set to be 305K, more assessments of the proposed NL correction method with such an internal BB one for all the detectors of FY-4B/GIIRS are provided in detail, particularly ΔBT values of both the interfered radiance within a wide band (640-1170 cm$^{-1}$) and the spectral radiance at each channel with the resolution of 0.625 cm$^{-1}$ at different referenced temperatures (i.e. 270K, 280K, 290K, 295K, 300K, 305K and 310K).

By using the proposed NL correction method, ΔBT values of the interfered radiance for all the detectors of FY-4B/GIIRS are provided as shown in Figure 10, which are almost less than 0.6K at different referenced temperatures between 270K and 310K. In general, the mean ΔBT values at the referenced temperatures above are around 0.3K except for at the relative lower temperature of 270K with its mean ΔBT about 0.4K. Particularly, for some detectors located near the marginal areas of FOV (i.e. 1$^{st}$, 16$^{th}$, 96$^{th}$, 113$^{th}$ and 128$^{th}$ as shown in Figure 5), parts of their ΔBT values are even larger than 0.5K.









**Figure 11. ΔBT of spectral radiance within each channel (with spectral resolution of 0.625 cm-1) for all the detectors of FY-4B/GIIRS before or after NL correction: (a) Before NL correction at 305K; (b) After NL correction at 305K; (c) After NL correction at 270K; (d) After NL correction at 310K.**

In addition, to assess the proposed NL correction method for the hyperspectral measurements from FY-4B/GIIRS in such a way which is identical with that for in-orbit radiometric calibration by using the onboard internal BB target, more analyzed ΔBT values under different conditions are plotted in Figure 11. As expected, the ΔBT values without NL correction are at least larger than 0.7K for both all the detectors and all the channels as shown in subfigure 11(a), which fully indicates the importance of NL correction for a GIIRS-like sensor with some high accurate requirements (i.e. usually better than 0.5K) of observations. Correspondingly, the ΔBT values with the proposed NL correction for both each detector and each channel are provided in subfigures 11(b)-(d) at different reference temperatures (305K, 270K and 310K), respectively. In particular, there are two thresholds represented by two translucent planes in black for the three subfigures above, where the smaller ones (i.e. 0.4K for subfigures 11(b) and 11(d) and 0.5K for subfigure 11(c)) refer to the maximal ΔBT for the valid spectral





range of 680-1130 cm$^{-1}$ while the larger ones (i.e. 0.7K for subfigures 11(b) and 11(d) and 0.8K for subfigure 11(c)) for the real spectral range of 640-1170 cm$^{-1}$. Obviously, for some marginal areas (for example less than 680 cm$^{-1}$ and more than 1130 cm$^{-1}$) of the observable spectrum, the ΔBT values behave significantly larger due to the relative lower of optical

efficiency of GIIRS. On the other hand, for the lower reference temperature (i.e. 270K compared with 305K and 310K), the ΔBT values are a bit larger, the main reason of which is the typical nonlinear relationship of measurement described in different radiometric units (namely between radiance and BT) within infrared band. Nevertheless, at the ordinary reference temperature of 305K, the mean ΔBT values for most detectors for all the valid channels within the valid spectral range of 680-1130 cm$^{-1}$ are usually around 0.2-0.3K, which is suitable for most common applications.

## 4. Discussion


In the proposed method, the basic roadmap of NL correction for a FTS (i.e. GIIRS) is clearly established, where the board band interfered radiance (namely interferogram) with respect to the external BB at different temperatures are selected to construct an overdetermined set of equations to calculate both the linear and NL coefficients, and the NL parameter $\mu$ independent of different working conditions (i.e. different temperatures for optical and mechanical components of sensor) is

derived further to implement the NL correction during a practical in-orbit calibration with the two-point method. Particularly, the NL parameter $\mu$ is regarded as a constant which is determined before launch and applied after launch. However, due to some possible causes (i.e. ice contamination) which are not totally understood (Guo and Feng, 2017), the SRF of GIIRS may be affected to a certain extent since it was launched. Apparently, it is inaccurate that the NL parameter $\mu$ of GIIRS is assumed to be currently identical with that of before launch. Therefore, some additional processings (i.e. in-orbit SRF

modification) are in needed before such a proposed method can be practical for implementation.

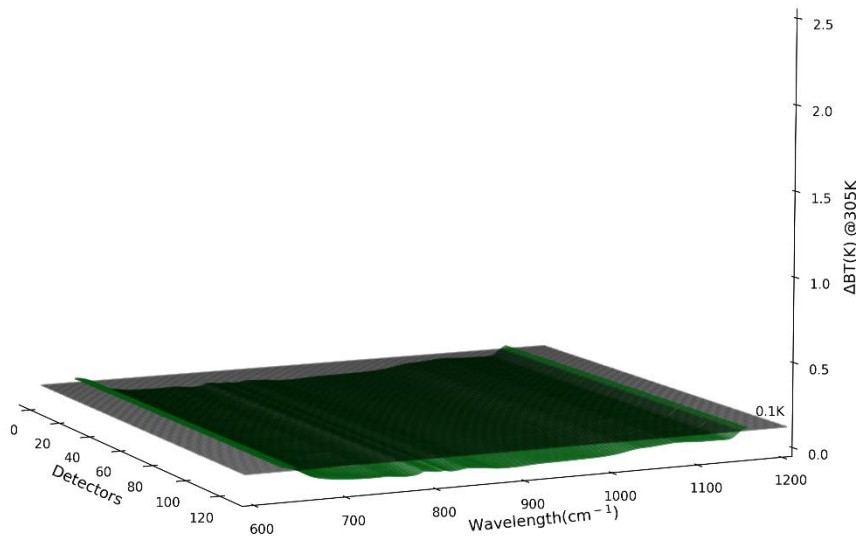





**Figure 12. ΔBT of spectral radiance for both all the detectors and all the channels of FY-4B/GIIRS after NL correction with the external BB target**

As shown in subfigure 11(b), although an internal BB target with its temperature of 305K is adopted to implement the two-
point calibration which is identical with that under in-orbit condition and the same temperature (305K) is also selected for reference to assess its NL correction performance, the practical ΔBT values are not satisfied to be around 0.2-0.3K with its maximal one less than 0.4K while their theoretical ones should approach zero. The main possible reasons come from the non-ideal characteristics of the adopted internal BB target with its emissivity much less than unit (i.e. 0.97-0.99 within the spectral range of 640-1170 cm$^{-1}$). In fact, under such a situation, the observed radiance from such an internal BB target by
detector is not merely from BB itself and some reflected radiation from its environmental components nearby must be considered with a certain compensation algorithm (Guo et al., 2021b). However, the estimated radiometric contribution will inevitably introduce some additional uncertainty of around 0.1-0.2K (Guo et al., 2021b), which finally causes the observable radiance from such an internal BB target is inaccurate enough. To valid such a conclusion above, some measurements from the external BB target with its temperature at 305K are chosen for calibration where no additional radiation are required to
be considered thanks to its low enough environment (i.e. generally less than 110K) in a vacuum chamber. As indicated in Figure 12, the distribution of ΔBT values under such a condition are almost less than 0.1K even for the real spectral range of 640-1170 cm$^{-1}$. It implies that the practical performance of the proposed NL correction method is partially dependent of the adopted internal BB target for calibration, which means the higher emissivity will produce the better NL correction.

Although the classical method of NL correction for an onboard FTS is widely applied in most similar sensors, the
determined parameter $b_2$ cannot be absolutely accurate, which at least depends on the determination of out-of-band spectrum which is assumed to be zero in practice. Therefore, the relationship between linear coefficient ($c_1$) and the parameter $b_2$ can be drawn from Eq.(20) as follows,

$$c_1 = \frac{I_r}{(2DN_{md}DN_{ma}+DN_{ma}{}^2)b_2+DN_{ma}} \approx \frac{I_r/(2DN_{md}DN_{ma}+DN_{ma}{}^2)}{b_2} , \tag{22}$$

In Eq.(22), the product between $c_1$ and $b_2$ approaches a constant, which means the relative errors for both $c_1$ and $b_2$ are
comparable. For example, 1% relative error of $b_2$ will cause around 1% relative error of $c_1$, the latter of which will introduce a bigger calibration error than that of the former. Such a conclusion can be partially validated by the results list in Table 4. Possibly, it is the main deficiency of the classical NL correction method for a FTS.

## 5. Conclusions

In this study, a new NL correction method for an onboard FTS is proposed. Totally, the NL parameter $\mu$ describing the
relationship between the linear and the NL coefficients is determined with some laboratory results before launch, which is utilized to carry out NL correction of an onboard FTS (i.e. GIIRS) together with the inaccurate enough linear coefficient by using the two-point calibration method. In particular, the NL parameter $\mu$ is confirmed to be independent of different





working conditions and can be in-orbit applied directly. Moreover, to overcome the inaccurate linear coefficient which is inevitably affected by NL response of sensor and impacted on the NL correction, an iteration algorithm is established to
make both the linear and the NL coefficients to be converged to their stable values with the relative errors less than 0.5% and 1% respectively, which is universally suitable for NL correction of both infrared and microwave sensors. By using the onboard internal BB which is identical with the in-orbit calibration, the final calibration accuracy for both all the detectors and all the channels with the proposed NL correction method is validated to be around 0.2-0.3K at an ordinary reference temperature of 305K. Significantly, in the classical method, the relative error of NL parameter immediately transmitting to
that of linear one in theory which will introduce some additional errors around 0.1-0.2K for the interfered radiance inevitably, no longer exists. Moreover, the adopted internal BB with the higher emissivity will produce the better NL correction performance in practice. The proposed NL correction method is scheduled for implementation to GIIRS onboard FY-4A satellite and its successor after modifying their possible SRF variations.

**Data availability:** All data generated or analyzed during this study are included in this article.

**Author contributions:** Qiang Guo and Xin Wang defined the project, Qiang Guo and Yuning Liu wrote most of the paper, Yuning Liu, Xin Wang and Wen Hui implemented data analysis and validation, respectively, and they thoroughly reviewed the manuscript.

**Competing interests:** The authors declare no conflict of interest.

**Acknowledgments:** We would like to thank Changpei Han of Shanghai Institute of Technical Physics, Chinese Academy of Sciences for kindly providing the laboratory results to this study.

**Financial support:** This research was funded by National Natural Science Foundations of China (42330110 and 41875037).

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
