# Peer review of "A New Non-linearity Correction Method for Spectrum from GIIRS onboard Fengyun-4 Satellites and its Preliminary Assessments"

_Atmospheric Measurement Techniques, 2023_

## Author Comment (AC1)

**Point by Point Response to RC1**

The reviews of our manuscript are thorough and well-considered. We would like to thank the reviewer for his/her careful reading and valuable comments to help us to improve this article. All the suggestions and comments from Referee 1 are addressed below point by point in bold text, followed by our responses in non-bold text. The corresponding revisions to the manuscript are marked in red. All updates to the original submission are tracked in the revised manuscript.

**This article discusses a new method for correcting spectral nonlinearity of GIIRS on Fengyun-4 satellite and its preliminary evaluation , to overcome the inaccurate linear coefficient which is inevitably affected by NL response of sensor and impacted on the NL correction, an iteration algorithm is established to make both the linear and the NL coefficients to be converged to their stable values with the relative errors less than 0.5% and 1% respectively, which is universally suitable for NL correction of both infrared and microwave sensors.**
**The following issues need to be considered:**

**Comment 01: The proposal of innovative points needs to be further summarized.**
**Response 01:** In this study, the NL correction is directly applied to the interfered broadband radiance observed by a spaceborne FTS (i.e. GIIRS). During prelaunch laboratory calibration, NL coefficients can be calculated by fitting the theoretical received radiance and the maximal DN at absolute ZPD at different temperatures by least square method. Finally, the NL parameter μ describing the relationship between the above linear and NL coefficients are determined, which is utilized to implement NL correction of such a FTS (i.e. GIIRS) together with the inaccurate enough linear coefficient by using the two-point calibration method. In addition, the NL parameter μ is almost independent of different working conditions and can be in-orbit applied directly. Moreover, to overcome the inaccurate linear coefficient which is inevitably affected by NL response of sensor and impacted on the NL correction, an iteration algorithm is established to make both the linear and the NL coefficients to be converged to their stable values respectively, which is universally suitable for NL correction of both infrared and microwave sensors.
The above contents have been supplemented in the original manuscript. Please refer to lines 522-534 in section 5 of the revised manuscript.

**Comment 02: The conclusion needs to provide prospects for further work.**
**Response 02:** In the further work, the adopted internal BB with the higher emissivity will produce the better NL correction performance in practice. The proposed NL correction method is scheduled for implementation to GIIRS onboard FY-4A satellite and its successor after modifying their possible SRF variations. Moreover, the real measurements from GIIRS after NL correction can be inter-calibrated with those of a reference sensor, i.e. IASI and CrIS to validate its calibration accuracy after NL correction.
The above contents have been supplemented in the original manuscript. Please refer to lines 541-544 in section 5 of the revised manuscript.

---

## Author Comment (AC2)

**Point by Point Response to RC2**

The reviews of our manuscript are thorough and well-considered. We would like to thank the reviewer for his/her careful reading and valuable comments to help us to improve this article. All the suggestions and comments from Referee 2 are addressed below point by point in bold text, followed by our responses in non-bold text. The corresponding revisions to the manuscript are marked in red. All updates to the original submission are tracked in the revised manuscript.

**As the first hyperspectral infrared sounder onboard geostationary platform, GIIRS's measurements will be significantly benefit to the local NWP prediction as well as temperature and humidity profile retrievals, which are mainly guaranteed by its high quality spectrum, particularly some nonlinearity correction (NL) processing upon its observations with enough accuracy. To overcome the shortcomings of the traditional NL one, a new approach dealing with the NL correction of GIIRS is proposed where both the NL parameter μ and an iterative algorithm are established with a better performance. In my opinion, such a paper can be accepted for publication before several minor issues are clarified.**

Comment 01: **Please supply the apodization characteristics of GIIRS measurements for both FY-4A and FY-4B satellites in Table 1.**

**Response 01:** This comment has been adopted by the authors. The apodization function is not applied to GIIRS for nether FY-4A nor FY-4B satellites. In addition, in order to make it convenient for users to do apodization processing, two channels of data are added to FY-4B/GIIRS L1 products on both sides of each band.

The supplements have been modified in Table 1 of the original manuscript. Please refer to Table 1 in lines 47-48 of section 1 of the revised manuscript.

Table 1. Main Specifications of LWIR and MWIR bands for GIIRS onboard FY-4A/B satellites

| Satellite | FY-4A | FY-4B |
|---|---|---|
| Spectral Range | LWIR: 700-1130 cm$^{-1}$
MWIR: 1650-2250 cm$^{-1}$ | LWIR: 680-1130 cm$^{-1}$
MWIR: 1650-2250 cm$^{-1}$ |
| Spectral Resolution | 0.625cm$^{-1}$ | 0.625cm$^{-1}$ |
| Spectral Channels | LWIR: 689  MWIR: 961 | LWIR: 721  MWIR: 961 |
| Number of Detectors | 128: 32×4 | 128: 16×8 |
| Spatial Resolution (@nadir) | LWIR/MWIR: 16 Km | LWIR/MWIR: 12 Km |
| Sensitivity (mW·m$^{-2}$·sr·cm$^{-2}$) | LWIR: 0.5-1.1  MWIR: 0.1-0.14 | LWIR: <0.5  MWIR: <0.1 |
| Radiometric Calibration accuracy | 1.5 K | 0.7 K |
| Spectral Calibration accuracy | 10 ppm | 10 ppm |
| Apodization characteristics | No apodization | No apodization |

**Comment 02: In table 2, the principles of NL correction for different sensors should be clarified more clearly.**

**Response 02:** This comment is helpful and has been adopted by the authors. The principle of NL correction for a hyperspectral infrared FTS is to evaluate and correct the NL of target spectrum according to its out-of-band artifacts in the low-frequency caused by NL. Meanwhile, the principles of NL correction for the wide-band infrared sensor and the microwave sensor are similar, measuring and correcting NL characteristics of a sensor during its calibration procedure, where the calculation of the linear and NL coefficients is mainly based on the mathematical form of calibration in radiance or BT with DNs measured by a sensor.

The supplements have been modified in Table 2 of the original manuscript. Please refer to Table 2 in lines 90-91 of section 1 of the revised manuscript.

**Table 2. Comparison of NL correction methods for different types of sensors.**

| Sensor Type | Hyperspectral Infrared FTS | Wide-band Infrared Sensor | Microwave Sensor |
|---|---|---|---|
| Principle | Evaluate and correct the NL of target spectrum according to its out-of-band artifacts in the low-frequency caused by NL | Measure NL characteristics of sensor and correct them in calibration procedure. Calculate the linear and NL coefficients mainly based on the mathematical form of calibration in radiance or BT with DNs measured by a sensor. | |
| Application | The interferogram is corrected by NL coefficient and then transferred into spectrum, which behaves linear relationship with radiance. | The NL coefficient is obtained with laboratory calibration and considered to be constant in-orbit, while the linear coefficient is achieved by two-point calibration method. | Both the linear and the NL coefficients are determined by using the NL parameter calculated during laboratory calibration as well as the linear coefficient calculated by two-point calibration method. |

**Comment 03: Please provide the physical meaning or explanation of NL parameter $\mu$ in the new method in detail.**

**Response 03:** This comment is good and has been adopted by the authors. The NL parameter $\mu$ describes the NL characteristic of a sensor itself. It denotes the relationship between the linear and NL coefficients obtained from the contribution of the linear and NL parts to the whole radiometric response of a sensor, representing the shape feature of the NL curve unrelated to radiance from targets, which is ordinarily independent of different working conditions of a sensor in theory.

The above contents have been supplemented in the original manuscript. Please refer to lines 212-215 in section 2.2.3 of the revised manuscript.

**Comment 04: In figure 8(b), the NL coefficients ($a_2$) for marginal detectors are generally smaller than those near the central of field-of-view, please analyze the possible reasons.**

**Response 04:** This comment is constructive and has been carefully considered by the authors. More analyzed results are provided in the revised manuscript. In figure 8(b), the values of NL coefficient ($a_2$) for marginal detectors are generally smaller than those near the central of field-of-view by about 50%, the main

reason of which is possibly caused by the overestimated linearity coefficients of the marginal ones due to the smaller incident radiation, making the estimated value of the linear part too large and further leading to the calculated one of the NL part much smaller than the actual one (namely the significant smaller NL coefficients).

The above contents have been supplemented in the original manuscript. Please refer to lines 352-359 in section 3.2.2 of the revised manuscript.

---

## Author Comment (AC3)

**Point by Point Response to CC1**

The reviews of our manuscript are thorough and well-considered. We would like to thank the reviewer for his/her careful reading and valuable comments to help us to improve this article. All the suggestions and comments from community comment 1 are addressed below point by point in bold text, followed by our responses in non-bold text. The corresponding revisions to the manuscript are marked in red. All updates to the original submission are tracked in the revised manuscript.

As the first hyperspectral infrared sounder onboard geostationary platform, GIIRS's measurements will be significantly benefit to the local NWP prediction as well as temperature and humidity profile retrievals, which are mainly guaranteed by its high quality spectrum, particularly some nonlinearity correction (NL) processing upon its observations with enough accuracy. To overcome the shortcomings of the traditional NL one, a new approach dealing with the NL correction of GIIRS is proposed where both the NL parameter  $\mu$  and an iterative algorithm are established with a better performance. In my opinion, such a paper can be accepted for publication before several minor issues are clarified.

Comment 01: The NL parameter  $\mu$  is originally proposed and applied in microwave sensors. Please supply some more detailed explanations about its feasibility for infrared ones (i.e. GIIRS).

**Response 01:** This comment is quite constructive and has been adopted by the authors. In fact, the basic mathematic expression of NL characteristics of a microwave sensor is fully identical with that of an infrared one (i.e. GIIRS), where calculations of both the linear and the NL coefficients are mainly based on the mathematical form of radiometric calibration in radiance or BT with DNs measured by a sensor. Thus, the NL parameter  $\mu$  in microwave sensors can be referenced for application in an infrared one. Moreover, the NL coefficients in infrared sensors are actually inconstant while the NL parameter  $\mu$  representing the relationship between the linear and NL coefficients is generally invariable, which is more suitable for description of the NL characteristics of an infrared sensor.

The supplements have been modified in the original manuscript. Please refer to lines 216-222 in section 2.2.3 of the revised manuscript.

**Comment 02: In section 4, three influencing factors, i.e. SRF variation under in-orbit condition, nonideal onboard BB source and the amplification effect of NL coefficient upon linearity one in the traditional method, are briefly discussed. It is recommended to add the corresponding subtitles to make these issues more clearly for readers.**

**Response 02:** This comment has been adopted by the authors. According to the three influencing factors, the section 4 is divided into three parts by subtitles, '4.1. SRF variation under in-orbit condition', '4.2. Non-ideal onboard BB source' and '4.3. Amplification effect of NL coefficient upon linearity one'.

The corresponding modifications have been made in section 4 of the original manuscript. Please refer to lines 481-520 in section 4 of the revised manuscript.

**Comment 03: In figure 2, three labelled information in parallelograms need to be given in a more accurate manner. For example, these parallelograms may be deleted directly.**

Response 03: This comment is helpful and has been adopted by the authors. Modifications have been made

in Figure 2 of the original manuscript. Please refer to lines 120-123 in section 2.2.1 of the revised manuscript.

Figure 2. The simple schematic diagram of Michelson interferometer.

---

## Author Comment (AC4)

**Point by Point Response to CC2**

In general, all the authors of this manuscript shall show their appreciations to the community comments from Dr. Gerald Turner (CC2) on amt-2023-242, some (i.e. Comment 01 and Comment 09) of which do help us to improve our paper. Here, point-by-point responses to these comments are provided as follows:

**General Comment:** The manuscript deals with the problem of nonlinear correction in hyperspectral infrared sounders of GIIRS. The authors attempt to provide a new nonlinear correction method based on the NL parameter estimation from the pre-launch radiometric calibration tests. However, the authors seem do not understand the cause of nonlinearity in infrared sounders, so the correction methods provided are not clear in logic. The final calibration results after nonlinear correction are not in accord with the calibration characteristics of similar instruments. In addition, the English writing and illustrations are terrible. I recommend that this manuscript be rejected for publication. I also recommend that the editor find other experts who are familiar with infrared sounder radiometric calibration to review this manuscript again. (e.g. Robert Knuteson, David Tobin, and Joe Taylor from University of Wisconsin–Madison, Dorothee Coppens from EUMETSAT, Laura Le Barbier from CNES).

**General Response:** It is a pity that Dr. Gerald Turner (CC2) provided a completely negative comment about this manuscript, the main reason of which is Dr. Gerald Turner believed that all the authors of amt-2023-242 **DO NOT** understand the cause of nonlinearity (NL) in infrared sounders. Doubtlessly, as pointed out by Y.Han (2018), NL of an infrared sensor (imager or sounder) is ordinarily dominated by the adopted detectors themselves (i.e. HgCdTe detectors for most spaceborne sensors). Some relevant statements or descriptions have been given in our manuscript, please refer to Lines 82-83. If the above-mentioned cause of NL which is also provided in our paper cannot be accepted by Dr. Gerald Turner, we have to say Dr. Gerald Turner know little about NL at least for infrared sensors. Otherwise, we **do recommend** Dr. Gerald Turner to reconsider such a negative comment about our manuscript made previously. As for the results with NL correction are not in accord with those from similar sensors, it is mainly based on different references for assessments. Moreover, this manuscript has been polished by an English native editor before submission without any terrible writing.

**Comment 01:** line 15~17 in abstract, "the NL parameter $\mu$ independent of different working conditions (namely the thermal fields from environmental components) can be achieved from laboratory results before launch and directly utilized for in-orbit calibration." The author needs to explain why the nonlinear coefficient is not affected by the temperature field. I don't find any discussion of this assertion in the paper.

**Response 01:** This comment is helpful for the authors to improve this manuscript. Here, we want to provide some more explanations about this issue. The $\mu$ proposed in our paper is NL parameter, describing the relationship between linearity coefficient ($a_1$) and NL coefficient ($a_2$), both of which vary with different working conditions (namely the thermal

fields from environmental components), and assumed to be almost constant at least against those of $a_1$ and $a_2$. In Line 347-352, the $56^{th}$ detector is selected as an example to explain why NL parameter $\mu$ is suitable for in-orbit NL correction and can be estimated using prelaunch testing results.

To make readers to understand more clearly, some additional analysis are supplemented in the revised manuscript, as list in the following Table 1. By using the prelaunch testing results, the main radiometric responsive characteristics (i.e. $a_1$, $a_2$ and $\mu$) of FY-4B/GIIRS under three working conditions are listed in Table 1 for some detectors located at two typical positions (namely marginal and central ones). In general, for FY-4B/GIIRS, its linear ($a_1$) and NL($a_2$) coefficients indeed vary significantly with different working conditions although temperatures for both detector operation and Aft optics remain stable, while the relative variations of $a_2$ are always larger than those of $a_1$ particularly for Cold condition. Moreover, the $\mu$ parameter established in the proposed NL correction method appears more stable than both $a_1$ and $a_2$ especially under both Normal and Cold conditions. This is the main technical foundation why the $\mu$ parameter is introduced in our method with an iterative algorithm to achieve the NL coefficient ($a_2$) with the higher accuracy.

The above contents have been supplemented in the original manuscript. Please refer to lines 359-373 in section 3.2.2 of the revised manuscript.

Table 1 Radiometric responsive characteristics of FY-4B/GIIRS under different working conditions for three detectors located at two typical positions

| Working condition and relative variation | | FOV-16 (marginal) | | | FOV-56 (central) | | | FOV-96 (marginal) | | |
|---|---|---|---|---|---|---|---|---|---|---|
| | | $a_1$ $(\times 10^{-2})$ | $a_2$ $(\times 10^{-7})$ | $\mu$ $(\times 10^{-4})$ | $a_1$ $(\times 10^{-2})$ | $a_2$ $(\times 10^{-7})$ | $\mu$ $(\times 10^{-4})$ | $a_1$ $(\times 10^{-2})$ | $a_2$ $(\times 10^{-7})$ | $\mu$ $(\times 10^{-4})$ |
| Hot | | 4.3897 | 4.6115 | 2.3932 | 4.2598 | 6.2569 | 3.4481 | 4.5365 | 4.0832 | 1.9841 |
| Normal | | 4.4029 | 4.5773 | 2.3612 | 4.2688 | 6.2198 | 3.4132 | 4.5599 | 4.0776 | 1.9611 |
| Cold | | 4.6140 | 4.9121 | 2.3073 | 4.4616 | 6.6329 | 3.3321 | 4.7546 | 4.3118 | 1.9074 |
| Mean value | | 4.4689 | 4.7003 | 2.3539 | 4.3301 | 6.3699 | 3.3978 | 4.6170 | 4.1575 | 1.9508 |
| Relative variation vs. mean value | Hot | 1.8% | 1.9% | 1.7% | 1.6% | 1.8% | 1.5% | 1.7% | 1.8% | 1.7% |
| | Normal | 1.5% | 2.6% | 0.3% | 1.4% | 2.4% | 0.5% | 1.2% | 1.9% | 0.5% |
| | Cold | 3.2% | 4.5% | 2.0% | 3.0% | 4.1% | 1.9% | 3.0% | 3.7% | 2.2% |

**Comment 02:** line 17~19, "to overcome the inaccurate linear coefficient from two-point calibration influencing the NL correction, an iteration algorithm is established to make both the linear and the NL coefficients to be converged to their stable values with the relative errors less than 0.5% and 1%" Nonlinear correction should be done before radiometric calibration, how to ensure the applicability of iteration coefficient? The nonlinear response of CrIS is only 0.13%, and the remaining 1% nonlinear coefficient is still too large.

**Response 02:** This question and the related comment are interesting. In fact, the aim of NL correction is to overcome the NL responsive characteristics of detector, which can be done either before radiometric calibration like CrIS or during radiometric calibration like GIIRS. No evidence is convinced that such a NL correction should be done before radiometric calibration in theory. If applicable, please provide the derivations in detail. Figure 3 in our manuscript shows the iterative algorithm flow of in-orbit NL correction, where the relative accuracies of both linearity and NL coefficients are mainly dominated by the pre-settable

parameter σ. At present, the relative accuracy of 1% for GIIRS NL correction is reasonable, which is comparable with its radiometric resolution (namely sensitivity) as well as the relative proportion of NL response. Actually, the accuracy of the proposed iterative algorithm can be increased with a smaller σ (i.e. 0.0001) to support the higher requirement such as 0.13% for CrIS.

**Comment 03:** line 21~23, "the final calibration accuracy for both all the detectors and all the channels with the proposed NL correction method is validated to be around 0.2-0.3K at an ordinary reference temperature of 305K." The radiometric calibration deviation after nonlinear correction should meet the calibration accuracy requirements at a series of blackbody temperatures, rather than just one temperature point, especially the temperature is close to the onboard blackbody temperature, and the deviation without nonlinear correction is inherently small, so this conclusion is not convincing.

**Response 03:** This is a good comment. In abstract, limited by the article space, only the BT difference at an ordinary reference temperature of 305K is provided to illustrate the preliminary performance of the proposed method. Figure 10 shows the more results for different detectors at different reference temperatures, including 270K, 280K, 290K, 295K, 300K, 305K and 310K.

**Comment 04:** line 23~25, "in the classical method, the relative error of NL parameter immediately transmitting to that of linear one in theory which will introduce some additional errors around 0.1-0.2K for the interfered radiance inevitably, no longer exists." This statement is ambiguous.

**Response 04:** The above statement is based on the analyzed results in Section 4 Discussion (particularly the derivation of Eq.22) as well as the related ones in Table 4. Please refer to the relevant contents.

**Comment 05:** line 25~26, "the adopted internal BB with the higher emissivity will produce the better NL correction performance in practice." Nonlinear response is a characteristic of the non-ideal IR sounder, independent of the emissivity of the blackbody. The authors need to provide clear verification evidence to explain why the use of high emissivity blackbodies will lead to better NL performance.

**Response 05:** This is a good comment and the basic statement of CC2 is right. Although NL response is a characteristic of a IR sounder independent of all the factors (including the emissivity of the adopted blackbody) outside sensor itself, when a BB is adopted for radiometric calibration in which the NL correction is implemented together as proposed in our method, the real incident radiance from such a non-ideal BB should be compensated due to the influence of its environmental components, which cannot be removed in theory. Therefore, in practice, the adopted internal BB with the higher emissivity will introduce less compensated radiance (please refer to Guo, et al., 2018, QJRMS, 147, 1562-1583) to produce the better NL correction performance. More detailed analysis have been provided in the 2$^{nd}$ paragraph of Section 4 Discussion of our original manuscript for reference.

**Comment 06:** line 44~45, "It was partially validated by both domestic and international users that the spectral and radiometric accuracies of the measured spectra from FY-4A/GIIRS V3 algorithm for L1 data show a well behaved performance for both LWIR and MWIR bands" The authors need to be honest and admit that GIIRS-A is not good due to spectral contamination, spectral calibration, and radiometric calibration. the cited paper only presents the result of a short period of time and does not represent long-term performance.

**Response 06:** Here, the authors have to remind Dr. Gerald Turner (CC2) to respect the historical status of FY-4A/GIIRS as the first GEO infrared sounder in worldwide, which suffered severe contamination in both LWIR and MWIR bands particularly for the latter one. Suffered from a lot of troubles (including contamination) upon measurements of FY-4A/GIIRS, its spectra after both spectral and radiometric calibrations can be acceptable since November 2019 for full utilization (i.e. L. Clarisse, et al., 2021, *Geophysical Research Letters*, 48, e2021GL093010), thanks to hard work of our team members. Therefore, many lessons from FY-4A/GIIRS should be learned in both instrument development and data processing. In fact, "a well behaved performance" here refers to "around 10 ppm of spectral calibration for both LWIR and MWIR bands and 1K of radiometric calibration for LWIR channels without contamination", which has been reported by the cited paper (Guo et al., 2021b) and further validated by other publicly oral presentation (oral.2.06.Theodore_itsc24) and poster (poster.7p.01.Burrows-Chris _itsc24) from ITSC-24 meeting in March 2023. Obviously, the period between November 2019 and March 2023 covers at least a 3-year period of time. Thus, such a challenge to long-term performance of FY-4A/GIIRS from Dr. Gerald Turner (CC2) is unreasonable.

**Comment 07:** line 46~47, "in order to increase the radiometric accuracy further, a new NL correction method which is aimed to carry out the NL processing of GIIRS is proposed in this article" The authors claimed in the article that they are committed to improve the radiation calibration accuracy of GIIRS, but in the end they do not give the actual application of nonlinear correction in GIIRS-A or GIIRS-B, just some pre-launch test results of GIIRS-B.

**Response 07:** This comment is incomprehensible. In fact, the aim of proposing a new NL correction method is to improve the radiometric accuracy of GIIRS-like sensor, which is still undergoing and has been partially validated by the pre-launch testing results of FY-4B/GIIRS. In the near future, such a method can be considered for implementation after much more evaluations with respect to some real measurements from GIIRS sensors onboard FY-4 satellites if applicable. The proposed NL correction method together with its preliminary results are reported firstly in this article.

**Comment 08:** line 52~54, "for the LWIR and MWIR, the detectors have larger NL contributions to be corrected against those of SWIR which are negligible small without correction (Qi et al., 2020; Zavyalov et al., 2011)." According to the cited paper, the authors here consider the GIIRS mid-wave band to be CrIS or HIRAS mid-wave band, but referring to the parameters in Table 1, the infrared semiconductor material of 3 to 6 μm does not show a high nonlinearity, and in fact the GIIRS mid-wave band is more similar to the shortwave band of CrIS. I have learned from Qi and Lee in the CMA that GIIRS's mid-wave should be short-middle wave band (SMWIR), as defined by GIFTS 20 years ago. Lee has acknowledged that GIIRS mid-wave does not express a strong nonlinear characteristic, which is determined by the properties of semiconductor material. The nonlinearity of mid-wave claimed by the authors requires some definite evidences. And in the results section of this paper, the author does not show any results of nonlinear correction in mid-wave band.

**Response 08:** This comment is ambiguous and the authors cannot understand what CC2's purpose is. In line 52-54, the authors merely states some basic characteristics for most FTS sensors, which are mainly quoted from two references, namely Qi et al., 2020 and Zavyalov et al., 2011. Here, the authors **DO NOT** emphasize whether the NL characteristics of GIIRS in LWIR and MWIR bands are strong or weak, while the NL corrections can be considered since the proposed method is suitable for an ordinary FTS regardless of strength or weakness for NL. In fact, the main purpose of this article is to establish a new NL correction method for a GIIRS-like sensor, which is partially validated by implementations in LWIR measurements prelaunch. More validations about MWIR band are still undergoing and reported in the following manuscripts.

**Comment 09:** line 60~61, "By looking for nonzero intensity in low frequency regions where the detector response is known to be zero, the final NL coefficient can be obtained (Chase, 1984)." misquotation! D. B. Chase's quote is that "The most reliable and straightforward method of evaluating detector nonlinearity is to look for nonzero intensity in a single beam spectrum in low frequency regions where the detector response is known to be zero." Thus, look for nonzero intensity in a spectrum is just a straightforward method to detect the presence or absence of nonlinearity, rather than to obtain nonlinear coefficients.

**Response 09:** This is good comment and has been adopted by the authors. In the revised manuscript, the above-mentioned misquotation is modified, namely "By looking for nonzero intensity in low frequency regions where the detector response is known to be zero, the final NL coefficient can be obtained, i.e. for AERI sensor (Knuteson et al., 2004b)".
The corresponding modifications have been made in section 1 of the original manuscript. Please refer to lines 58-59 in section 1 of the revised manuscript.

**Comment 10:** line 63, "HIRAS (Qi et al., 2020; Wu et al., 2020)," misquotation! I have discussed with Qi and Wu in many meetings, such as GSICS、ITSC, that CMA HIRAS does not adopt the nonlinear correction method of UW-SSEC, because HIRAS nonlinear response is different with CrIS, and the authors need to know exactly how your CMA colleagues are doing.

**Response 10:** This is **NOT** a misquotation. Here, some relevant contents in the two cited papers are provided as follows, namely
Qi et al., 2020: "In the radiometric calibration of the FY-3D HIRAS measurements, the NL levels of the LW and MW detectors are high enough to carry out an NL correction to the uncalibrated spectra using the methods summarized in Wu et al., 2020"
Wu et al., 2020: "The NL correction algorithm operationally used for FY-3D HIRAS is an algorithmic approach first developed by the Space Science and Engineering Center at the

University of Wisconsin–Madison (UW-SSEC), in which only the quadratic NL term is considered."

According to the above cited papers, **HIRAS does adopt the NL correction method of UW-SSEC**. Also, the authors wonder whether or not CC2 has discussed with Qi and Wu about this issue!

**Comment 11:** line 64~65, The authors seem do not understand the nonlinear correction method of CrIS. CrIS definitely uses the UW-SSEC method, which is not a new method either in TVAC test or in orbit calibration. The corrected coefficients in orbit are only a little tuning on the basis of the pre-launch coefficients. In fact, CrIS detector has a well response linearity (>99.8%) and the post-launch fine tuning is to adapt the spacecraft environment change, rather than re-derived a set of new coefficients.

**Response 11:** As for the NL correction method of CrIS, all the cited contents are totally from such a classical reference (Han, 2018), where three methods for its NL correction are introduced separately. As reported, "The three methods working together overcome the shortages of these methods". Anyway, in our manuscript, the authors **DO NOT** claim that CrIS re-derived a set of new coefficients for NL correction. So, the above comment is without merit.

**Comment 12:** line 71~72, "it is in need to determine and correct the NL response during calibration, particularly for the quadratic contribution of NL." And in line 82~83, "The NL principle of GIIRS is essentially the same as that of the traditional broad band sensors, except that the band (LWIR and MWIR) of GIIRS is much wider." If the authors understand Chase's analysis, they should know that the infrared imagers and the infrared sounders have different nonlinear characteristics.

**Response 12:** This comment is of prejudice and Dr. Gerald Turner (CC2) has made it without reading the whole article. Actually, since the main dominated factor influencing the NL characteristics of a sensor (imager or sounder) comes from adopted detectors, the NL principle of a FTS (i.e. GIIRS) is essentially the same as that of the traditional broad band sensors. At the same time, due to the inevitable phase errors during the sampling procedure of a FTS, phase correction should be done or phase information needs to be known ahead of time before implementing such a NL correction. In subsection 2.2.2, subsample location alignment is carried out to eliminate or at least decrease the influence of phase error upon NL correction. Thus, there is no obvious difference in NL characteristics between imager and sounder except that some phase correction is in need before implementing NL correction of a FTS.

**Comment 13:** line 75~78, the author needs to discuss how the instrument temperature field affects the detector nonlinearity. It is unreasonable to confuse the nonlinearity of microwave instruments and infrared sounders, since they utilize different detectors and have different response nonlinearities.

**Response 13:** This comment is reasonable. For a spaceborne IR sensor (i.e. imager or sounder), the adopted detectors are mostly photonic ones, for example made of HgCdTe materials, the radiometric responses of which are at least dominated by their background

radiation from the instrumental temperature fields. As one component of the whole radiometric response, NL response of such a photonic detector is certainly affected background radiation. Please refer to Guo, 2021a for more relevant information. In addition, although different detectors are utilized in infrared and microwave sensors, the mathematic expressions of NL response for the two types of sensors are almost identical, which makes it possible to overcome such a NL correction in a similar way or at least for reference mutually. In our article, the NL parameter $\mu$ which is originally defined for microwave sensor is introduced and more improvements (i.e. an iterative algorithm) are established for GIIRS to achieve the better results.

**Comment 14:** line 86, "spectral response function (SRF)" The authors need to know that the spectral response function (SRF) or ISRF is the response of the instrument to a beam of monochromatic light. The term has been defined by D. Siméoni et. al. for IR-FTS like IASI in 1990s. Eq.(5) is correct to use SRF, in an ideal or well spectrally calibrated FTS without apodization, SRF should be a sine cardinal function, rather than the sounder responsivity in Figure 6. And in line 289~290, "SRF of a sensor (i.e. GIIRS) generally refers to the ratio of the received radiation relative to the incident radiation at each wavenumber." This is not a definition of the spectral response function of equation (5), but a definition of the spectral responsivity of the instrument. Spectral response function and spectral responsivity have very different meanings.

**Response 14:** This comment refers to the definitions of some terminologies. As indicated previously, since a FTS is essentially a broad band sensor to measure the interfered radiance with different optical path differences (namely interferogram), the terminology of spectral response function (SRF) should be identical with its original one which represents the spectral responsivity of sensor and can be mainly dominated by characteristics of both optics and detectors. However, due to non-ideal sampling upon interferogram from targets, the response of a FTS to a beam of monochromatic light is a sine cardinal function rather than an ideal impulse function, which is traditionally called instrumental line shape (ILS) function. Therefore, the authors will continue to utilize the related terminologies in the revised manuscript.

**Comment 15:** line 90~91, "which provide a new and more accurate way for in-orbit NL correction for both infrared and microwave sensors in theory." The author should give some examples of the application of nonlinear coefficients in microwave sounders. I think discussing microwave nonlinear correction in this article deviates from the topic of the paper.

**Response 15:** This comment is ambiguous. In line 90-91 of our article, we just claim that the proposed NL correction method is theoretically more accurate, which can be applied in both infrared and microwave sensors since the mathematic expressions of NL characteristics for the two types of sensors are identical with each other. Thus, there is unnecessary to provide some applications in microwave one as pointed out by CC2 later.

**Comment 16:** Table 2 is dispensable.

**Response 16:** Table 2 summarizes the main NL correction methods for different types of sensors, which helps readers to understand the proposed one more clearly. So, table 2 is reserved in the revised paper.

**Comment 17:** line 103~104, "which means the alternating current (AC) component of target radiance is retained." In FTS, AC component generally refers to the interferogram without non-interference term, that is DC term, so $I$ is an optical interferogram rather than a radiation spectrum, but the authors have considered the equation (1) is based on the radiance of the infrared imager, so this is wrong. And the ignoring of a0 is too arbitrary.

**Response 17:** The authors recommend Dr. Gerald Turner (CC2) to have a deep understanding of the operating principles of a FTS (i.e. GIIRS). As indicated in Response 14, a FTS is essentially a broad band sensor to measure the interfered radiance with different optical path differences, which is exactly satisfied with the Fourier transform of radiance from targets in mathematics. This is why the outcomes of a FTS are usually called interferogram which is indeed the interfered radiance from targets. Viewing from the aspect of radiation, two interferograms (or called interfered radiances) from target and deep space can be subtracted mutually. Moreover, viewing from the aspect of Fourier transform, the two interferograms can also be subtracted, results of which equals to Fourier transform of subtracted radiance between target and deep space. Particularly, the interfered radiance or called interferogram at absolute ZPD location is equal to twice of incident radiance from target. Please refer to subsection 2.2.1. Therefore, equation (1) is completely right.

**Comment 18:** line 110~111, "In the step of NL coefficient extraction, after convolving BB radiance with sensor's SRF, the theoretical interfered radiance (namely interferogram) received by GIIRS can be obtained." The convolution of the blackbody spectrum with SRF is still a spectrum, not an interferogram.

**Response 18:** The relevant explanations about interfered radiance as well as interferogram have been provided in Response 17.

**Comment 19:** line 113~114, "during laboratory calibration, NL coefficients (a2) can be calculated by fitting the DN with the radiance at different temperatures (180K, …, 320K) by least square method." DN is interferogram, while radiance is spectrum. Without Fourier transform, the interferogram cannot be fitted with radiance.

**Response 19:** Again, the authors want to emphasize that only interfered radiance at the absolute ZPD location is selected for radiometric calibration together with NL correction, and is equal to twice of net radiance from BB target. No Fourier transform is in need here.

**Comment 20:** line 116~119, this description is ambiguous.

**Response 20:** These descriptions in line 116-119 are to show the in-orbit application of NL correction with the determined NL parameter μ prelaunch as well as the proposed iterative algorithm to achieve linearity and NL coefficients with more accuracy. Please refer to Figure 1 for easy understanding.

**Comment 21:** In Figure 2, the BB radiance has been modulated as an interferogram in temporal space, rather than a spectrum, and therefore cannot be convolved with SRF in frequency space. The signal received by the detector is interferogram rather than radiance.

**Response 21:** Again, the authors claim that the interferogram of a FTS (i.e. GIIRS) is essentially the interfered radiance with different optical path differences from target, the unit of which is same as that of traditional radiance. Once the corresponding optical path difference and phase error are known, the interfered radiance can be calculated accurately. In our study, the interfered radiance (or called interferogram) at absolute ZPD (namely without influence of phase error) is utilized for radiometric calibration together with NL correction within the whole responsive band (i.e. LWIR) of a FTS, not merely for a single channel after Fourier transform.

**Comment 22:** Section 2.2.2 Subsample location alignment, the author's so-called "ZPD detection method" is not a new approach, which is just the "zero padding" commonly used in Fourier spectrum analysis. The "zero padding" method for spaceborne IR-FTS calibration has been introduced by Bob Knuteson et. al. in the 20th IIPS conference in 2004. In GIFTS spectral calibration, Bob et. al. utilizes zero padding in spectral space combined "double FFT" described in (Han, 2018) to get the real maximum optical path difference (MPD), not ZPD, of the off-axis interferogram, so the frequency shift in off-axis spectrum could be corrected, and the spectral sample is approximately equal to the reciprocal of the 2*MPD. Bob had realized the zero padding method is time-consuming in data processing, so an equivalent convolution method proposed by J. Genest and P. Tremblay was replaced. The convolution method is high-efficiency, and has been used spectral calibration of ABB Bomem's interferometers, like Aura/TES, ACE-FTS, IASI, CrIS. CMA HIRAS maybe use the same method as CrIS described by Qi. Therefore, I recommend that authors learn some history of the spaceborne FTS and make respect for the work that has been done.

**Response 22:** First of all, the authors shall show their appreciations for Dr. Gerald Turner (CC2) for providing some useful information about relevant works which have been on spectrum processing to a spaceborne FTS. In fact, the contents in subsection 2.2.2 about subsample location alignment are firstly reported in one reference (Guo, et al., 2021b) and aim to realize the absolute ZPD detection, where "zero padding" is adopted to achieve the subsampling misalignment due to phase error during sampling. Meanwhile, as indicated by Dr. Gerald Turner above, the "zero padding" method for an onboard IR FTS is mainly to extract the real MPD of some off-axis interferograms. Obviously, the same "zero padding" processing method can be used for different tasks. Thus, there is unnecessary to doubt the available method given in subsection 2.2.2.

**Comment 23:** line 179~183, this section has little to do with the topic of this paper. Table 3 also does not give some useful information to readers.

**Response 23:** This comment indicates that Dr. Gerald Turner (CC2) does not understand the proposed NL correction method to a great extent. Due to some inevitable phase errors introduced during sampling procedure of a FTS (i.e. GIIRS), the subsample location alignment is adopted to achieve the real phase errors to calculate the interfered radiance (namely interferogram) at the location of absolute ZPD, which can be done according to

equations (6-9). The relevant contents are given in line 179-183, which help readers to understand more easily. As examples listed in Table 3, the original and aligned DNs at the location of ZPD are provided for comparison, the difference of which is non-negligible. These results in Table 3 validate the necessity of subsample location alignment for the proposed NL correction method.

**Comment 24:** line 186~188 and Eq. (10), In theory, the interferogram value at ZPD characterize the integrated energy of blackbody over the spectral response range. But real FTS has some modulation degree less than unit of ideal FTS, so the value at ZPD is not necessarily correlated with nonlinearity. As far as the interferogram signal is concerned, the nonlinearity occurs at each OPD position of the interferogram, not only at ZPD, which is why the nonlinearity of FTS is different from that of the imager. In Eq(10), the rationality of using second-order polynomials to fit ideal radiation needs to be discussed, otherwise it will be too arbitrary.

**Response 24:** This comment is interesting. In theory, the modulation degree of a real FTS will influence the amplitude of interfered radiance (namely interferogram) within the whole spectral response range, which equals to the attenuation of interference optics (including both stationary and moving mirrors) of a FTS and can be considered in radiometric calibration. It is true that NL occurs at each OPD position of the interferogram, where different interferograms at ZPDs when viewing BB at different temperatures are utilized to estimate NL coefficient thanks to the definite interferogram at the absolute ZPD position given by equation (3). Moreover, since NL characteristics of an infrared FTS are mainly dominated by the adopted detectors, it is rational to use $2^{nd}$ polynomials for NL description.

**Comment 25:** line 194~198, The authors need to explain why the off-axis angle changes with the direction of the moving mirror. That's a weird statement. According to the FTS optical path layout, the off-axis angle is constant in instruments like IASI and CrIS. If the alignment of the moving mirror and the fixed mirror change, it means that the modulation degree of the instrument also changed, and the model of equation (10) is not valid. Description in line 196~198 is not correct, off-axis correction is to correct the shifted spectrum and more important the ISRF. The spectral frequency shift is originated from the off-axis spectrum using the spectral abscissa calculated by the sample interval of the on-axis interferogram. If we get the real OPD of the off-axis interferogram, then we can get the true abscissa without any frequency shift, but with different spectral scales or spectral sample intervals.

**Response 25:** This comment is interesting. Actually, the off-axis angle corresponding to an individual detector for an ideal FTS is independent of the direction of the moving mirror. However, after discussing with some experts from the vendor of GIIRS (SITP/CAS), they told us that there is slight differences in depth of parallelism between the moving and the stationary mirrors for different directions of the moving one, which causes different OPDs even for the same detector with respect to different sweep directions. Thus, such slight differences of OPDs can be equivalent to different off-axis angle coefficient (cosine value of off-axis angle) for different directions. It should be emphasized that such differences are slight around less than $2\times10^{-6}$.

**Comment 26:** Figure 3, In the flow chart, the updated coefficient a1^0 is the average of the original two coefficients $a_1^0=(a_1^0+a_1^1)/2$. That is so weird. The author needs to explain the rationality of this step.

**Response 26:** This comment is interesting. As indicated in figure 3, $a_1^0$ and $a_1^1$ are two rough estimated values of linearity coefficients without/with influence of NL effects, respectively. Regardless the positive or negative of NL coefficients, for the two estimated values, one is underestimated while the other is overestimated. Therefore, in the proposed iterative algorithm, the arithmetic mean of the two values are used to update $a_1^0$ coefficient for the following iterations.

**Comment 27:** Section 2.4, it is not necessary for the author to derive the UWM-SSEC method in Sec. 2.4 again. And in fact, CrIS nonlinear correction is not done on the DN values.

**Response 27:** This comment is constructive. However, to make readers to understand the classical NL correction method more directly, some simplified and important steps are given in section 2.4. In addition, no words here are claimed whether or not NL correction of CrIS is done on DN values.

**Comment 28:** line 353~355, "Similarly, compared with detectors near the central positions of FOV, the NL parameters ($\mu$) of the marginal ones are apparently underestimated by around 50% against those of central ones, which is also mainly induced by their bigger linearity coefficients" Generally speaking, the light intensity near the FTS optical axis is stronger than that at the edge of the field of view, which is common sense. The authors mistake it for a large linear coefficient, so the subsequent nonlinear correction results are not reliable.

**Response 28:** This comment is of prejudice without careful consideration. As indicated the words in line 353-355 of our manuscript, the NL parameter ($\mu$) of the marginal detectors are apparently underestimated against those of central ones. According to the definition of $\mu$, it is mainly induced by the bigger linearity coefficients ($a_1$) of marginal detectors. It is quite true that the less light intensity arrives the off-axis area of a FTS and further causes the lower responsivity of corresponding detector, which can be represented by the bigger linearity coefficient according to the basic radiometric calibration equation (Eq.1). Therefore, the analysis in line 353-355 are of course correct without any mistake, and the related comment from Dr. Gerald Turner (CC2), "the subsequent nonlinear correction results are not reliable" is entirely baseless.

**Comment 29:** line 355~356, "due to the relative lower optical efficiency at the locations near the marginal areas of FOV," The lower intensity at the edge of the field of view is due to off-axis effects rather than low optical efficiency.

**Response 29:** This comment is surprising. In fact, detectors near the marginal areas (notes: it indeed belongs to off-axis areas) of FOV receiving less light intensity is equivalent to the lower optical efficiency against those near the central ones. For a FTS, the off-axis effects mainly cause different OPDs.

**Comment 30:** line 357~358, "It implies that the radiometric responsivities of the marginal detectors are generally lower, which can further lead to the smaller NL parameters ($\mu$) even for the same detectors." the detector responsivity does not decrease with its distribution location.

**Response 30:** This comment is impracticable in spite of reasonable to some extent. It is true that detector responsivity does not decrease with it location. However, for detectors near marginal area of FOV, they receive less light intensity against those near central one when viewing the uniform targets, which is equivalent to a lower radiometric responsivity and can be represented by a bigger linearity coefficient. Here, the high or low responsivity of detector is relative and equivalent.

**Comment 31:** In Table 4, If the nonlinearity is corrected well enough, the coefficients can be applied to different observation scenes and do not vary with the energy of the observation targets, because nonlinearity is an inherent property of a non-ideal FTS. As can be seen from the table 4, both linear and nonlinear coefficients are variable, so it can be inferred that this new method is invalid and these coefficients have no practical values in atmospheric radiance calibration. What is strange is that the nonlinear coefficients obtained by the author using the UWM-SSEC method are also variable, which is completely different from the actual application of CrIS, so I think the author does not understand the UWM-SSEC method.

**Response 31:** This comment and the relevant statements are imprudent without considering the practical situations of GIIRS. The NL correction results listed in Table 4 mainly simulate the in-orbit application by using measurements from the onboard single internal BB target. Limited by the non-ideal BB with the relative emissivity within the whole spectral band (including LWIR and MWIR, around 0.95~0.99 at different wavelengths), the real incident radiance from internal BB cannot be calculated accurately due to some inevitable radiance reflected from its environmental components, which causes the variable linearity and NL coefficients for all the three methods. This is why the authors claim the onboard BB targets with higher emissivity can provide the more reliable NL correction. **The related explanations are also provided in Response 05**. Meanwhile, similar situations do not occur for CrIS thanks to its perfect onboard BB with its emissivity around 0.99. Nevertheless, the ΔBT values with the proposed method for different temperatures of adopted internal BB are generally stable without modifying the reflected radiance contribution from its environmental components, which partially validates the feasibility of the proposed NL correction method against those of the other two ones.

**Comment 32:** line 420~421, "Such results imply that the derived $b2$ values from the classical method are inaccurate." This conclusion is absurd. Maybe the authors do not understand the UWM-SSEC method.

**Response 32:** This is comment is irresponsible. These words in line 420-421 merely describe a fact that the derived $b_2$ is inaccurate, which is not certainly caused by the classical method itself or maybe some additional modifications should be done with respect to the practical situations of GIIRS. In fact, the authors **DO NOT** believe that the UWM-SSEC method can be directly applied to a spaceborne FTS (i.e. GIIRS) without any corresponding

modifications. The implementation of the classical method done by the authors is completely based on the descriptions in this reference (Han, 2018).

**Comment 33:** line 424~426, "From the perspective of NL correction, the NL characteristics of GIIRS are underestimated by the classical method, …" According to the height of the geostationary satellite and the spatial resolution of GIIRS pixels, it can be estimated that the pixel size of the GIIRS detector is not too large, about 100 μm, which is much smaller than CrIS pixel with a diameter of about 1 mm. Therefore, even though the performance of Chinese infrared detectors may not be comparable with that of CrIS, the nonlinear response of the GIIRS long-wave infrared HgCdTe detector should not be greater than CrIS. I think the UWM-SSEC method can easily correct the nonlinear bias of GIIRS, but the authors don't seem to really understand this method.

**Response 33:** The authors have to admit that Dr. Gerald Turner (CC2) is extremely conceited. As indicated by CC2, the greater NL characteristics of GIIRS are more difficult to be overcome totally against those relative smaller ones of CrIS. Again, we shall emphasize that it is impossible the UWM-SSEC method can be directly applied to a spaceborne FTS without some necessary corresponding modifications. GIIRS is not a copy of CrIS to be equipped on Chinese FY-4 satellite.

**Comment 34:** Figure 10 is worthless because people are more concerned with the spectral distribution of the radiometric calibration deviation of the infrared sounder than with the integrated energy.

**Response 34:** This comment is arbitrary. To illustrate the real performances of the proposed NL correction method for different detectors at different reference temperatures, figure 10 provides the corresponding ΔBT values of the integrated radiance for comparison by utilizing the proposed method. Therefore, figure 10 is necessary to be remained in the revised paper.

**Comment 35:** From Figure 11 it can be confidently confirmed that the radiometric calibration failed, let alone the non-linear correction. According to pre-launch radiometric calibration tests such as IASI and CrIS, the BT deviation of the blackbody calibrated spectra should not have significant spectral characteristics, and its mean value should be a straight line trend along the spectral abscissa. In Figure 11, The BT deviation still remains the characteristics of the instrument's spectral responsivity. It means that the calibration does not eliminate the instrument effects, thereby defeating the purpose of radiometric calibration.

**Response 35:** The comment is irresponsible from a technical perspective or maybe Dr. Gerald Turner (CC2) does know little about radiometric calibration of an infrared sensor. As shown in equation 1, the outcomes (DN) of a sensor will be calibrated according to the incident radiance which should be known ahead of time. Here, the physical quantity to be calibrated is radiance rather than BT. Thus, in this sentence of "its mean value should be a straight line trend along the spectral abscissa" given by CC2, the mean value should be referred to radiance instead of BT, while both ΔBT and delta radiance are not proportional within infrared spectral range. Meanwhile, the influence of NL upon smaller incident

radiance (i.e. measurements at the marginal position of SRF for a FTS) is much more than those of bigger ones (i.e. those near the central position of the whole spectral band). Here, the ΔBT after NL correction and its SRF of 56th detector of FY-4B/GIIRS are shown for comparison as in the following figures (a) and (b), respectively. Clearly, the characteristics of detector's SRF is almost eliminated particularly for main spectral region (i.e. 700-1000cm$^{-1}$). Therefore, the results of figure 11 are relative reasonable doubtlessly.

[Figure]

(a) ΔBT after NL correction for 56th detector          (b) SRF of 56th detector

**Comment 36:** line 443~445. As for in-orbit radiometric calibration, people prefer to see the calibrated spectra and the calibrated deviation of the real atmospheric radiance, rather than the onboard blackbody radiance. Actually, the onboard blackbody should serve as the reference for atmospheric radiation calibration.

**Response 36:** This comment is reasonable. However, since the proposed NL method is merely established with some preliminary validations by using the prelaunch BB calibration results, more detailed analysis are in need further under in-orbit condition. The aim of this manuscript is to introduce such a new NL correction method with a limited examples to evaluate its feasibility.

---

## Referee Report (RR1)

**Review of amt-2023-242**

GENERAL

This is detailed and invaluable work and digs deep into the calibration problems associated with orbiting FTS radiometers. Some questions that arise from the easy standpoint of someone who doesn't have to do it are given in the following detailed commentary section.

COMMENTARY

line: comment

14-19: Note that laboratory data are almost always taken with the gases at thermal equilibrium. In reality, the nonlinearity of the atmosphere means that temperatures and line shapes are not at LTE. See *Meteorology* **2023**, *2*(4), 445-463; https://doi.org/10.3390/meteorology2040026 and references therein.

45-55: Self-absorption in many atmospheric rovibrational transitions means that the wings are relatively more important than in an equilibrated laboratory sample. In the real atmosphere with non-LTE, that may pose problems for satellite FTS retrievals.

119 et seq: I have assumed that the equations in Section 2.2.1 are correct.

192-200: I note that J W Brault developed an interferometer wherein only the 'perfect' configurations were selected. The improved signal-to noise was such that commercial manufacturers adopted it (it was not patented). Of course, that is irrelevant for interferometers already in orbit - or can software be developed?

291-305: How does non-LTE in the atmosphere affect this?

323-325: Again, how does non-LTE play in this problem?